# Gestational Dysfunction-Driven Diets and Probiotic Supplementation Correlate with the Profile of Allergen-Specific Antibodies in the Serum of Allergy Sufferers

**DOI:** 10.3390/nu12082381

**Published:** 2020-08-09

**Authors:** Anna Maria Ogrodowczyk, Magdalena Zakrzewska, Ewa Romaszko, Barbara Wróblewska

**Affiliations:** 1Department of Immunology and Food Microbiology, Institute of Animal Reproduction and Food Research of Polish Academy of Sciences, Tuwima 10, 10-748 Olsztyn, Poland; 2The Center for Pulmonary Diseases, Allergy Outpatient Clinic, Jagiellońska 78, 10-357 Olsztyn, Poland; madazak1@interia.pl; 3Non-Public Health Care Clinic “ATARAX”, 1 Maja 3, 10-117 Olsztyn, Poland; ewaroma@mp.pl

**Keywords:** gestational dysfunction-driven diets, probiotics supplementation, foetal programming, allergy prevalence, specific IgE-antibodies profile, allergic patterns, artificial self-organizing neural network

## Abstract

Background: Maternal diet has significant effects on development of childhood atopic disease and hypersensitivity development. However, the gestational dysfunctions demanding special diets are becoming a widespread phenomenon, their immunological implications can be manifested in the profile of antibodies in the offspring’s serum. Methods: 153 allergic and 150 healthy individuals were diagnosed for allergy using specific antibody and cytokine immunoassay tests. The medical history of subjects along with mothers’ course of pregnancy was completed by allergologist’s anamnesis. A self-organizing neural network and multivariate analyses to complex data and pick basic interactions were used. Results: Two significant explanatory modules were determined. The first was formed by gestational diabetic and cholestatic diet, infant formula feeding type, probiotic supplementation and its BMI index, moderate IgE, increased IgG levels of antibodies and single or poly-food allergy type (7 clusters). The second was formed by gestational vegan/vegetarian and elimination diet, maternal probiotic supplementation, sex, high IgE total antibodies and food and mixed poly-allergy to aero- and food-origin allergens (19 clusters). Conclusions: Significant associations were observed between special gestational diet intake underlying foetal programming and the mechanisms of childhood allergy. The novelty is the positive association between diabetic and cholestatic diet intake and IgE/IgG-mediated food hypersensitivity.

## 1. Introduction

The human immune system is immensely complex but is not always able to develop tolerance to food allergens. The drawback of developing an immune system that may recognize and respond to infections is the potential for hypersensitivity reactions. These manifest as allergic responses to environmental agents among others. Despite the intensive advancement of knowledge in the area of aetiology and progression of allergic disease, this problem affects an increasing number of people. According to the World Health Organization [1], it is currently the third most commonly diagnosed chronic disease and food allergy rates are now higher than ever before. According to the European Academy of Allergy and Clinical Immunology (EAACI), more than 150 million Europeans suffer from chronic allergic diseases as per the current scenario and half of the entire EU population is foreseen to be affected by 2025 [2]. It has been estimated that the immune system of about half of newborns is not able to develop tolerance to food-origin proteins [3]. The global food allergy diagnostics and therapeutics market size was valued at 2.69 billion USD in 2018 and is expected to grow by 7.4% over the next 10 years, whereas the global allergy immunotherapies market is expected to reach 3.63 billion USD by the end of 2025, growing by 10.8% during 2019–2025 [4]. According to a report published in 2018 based on children less than 5 years old, the prevalence of challenge-proven food allergy has been reported to be 4% in the UK, 3.6% in Denmark, 6.8% in Norway [3], and in Poland in 2016, it was 8% [5]. The discrepancies and underestimation in those results are always difficult to interpret, given that allergies can be manifested immediately but can also give a delayed reaction.

The consequence of the progression of allergic diseases including food allergy is a number of possible health complications and the well characterised allergic march as a consequence of skin and epithelial barrier dysfunction [6,7]. However, the latest research mainly focuses on demonstrating possible links between allergic diseases and the development of subsequent eating disorders, psychosocial drawback [8], and inflammatory autoimmune diseases with humoral regulation [9]. Some of those diseases, because of commonalities in terms of susceptibility loci, genetic pathways, and genomic regulatory sites, may also explain the dramatic increase in the coincidence of mentioned disorders that are highly heritable.

Properly selected immunotherapy plays an important role and might be conducted through induction, enhancement or suppression of an immune response on different levels, including gene expression and its epigenetic activity modulation [10,11]. One of the most promising types of therapy is based on nutrigenomics. The role of foetal programming in the prevention and therapy of various inflammatory and atopic diseases are also the widely discussed randomized controlled trials on the influence of dietary intervention on epigenetic mechanisms in children who suffered from cow’s milk allergy that have already done [12] Maternal dietary patterns may prevent or reduce hypersensitivity with significant reduction especially in IgG-mediated diseases [13]. A successful protocol for IgE-mediated reactions still remains a challenge, but there have been several studies published describing correlations between maternal dietary components that works preventively through epigenetic activation of mechanisms against asthma, wheezing and allergic rhinitis [14]. Nevertheless, over the last 10 years, based on retrospective studies and precise maternal and children diet and lifestyle analyses, an earlier and a more severe failure of oral tolerance during the first year of life and the likelihood of outgrowing food allergies decreased, whereas the severity of diseases increased [12,15,16]. That forces changes in the approach to conducting therapy and stimulation to analyses of factors inducing the foetal immune system in prenatal time.

Diet remains one of the dominant environmental factors that has an influence on the immune system development and function and, thus, it affects many aspects of health and disease risk [17]. Some meta-analyses of data showed that maternal probiotic and fish oil supplementation may reduce risk of eczema and allergic sensitization to food, but at the same time, emphasized that the risk of bias was high (from 25–48%) in tested studies [18]. Based on that meta-data analysis, it was found that results regarding other gestational dietary exposures, including prebiotic supplements, maternal allergenic food avoidance, and vitamin, mineral, fruit, and vegetable intake, were inconclusive or inconsistent.

Not only was the role of gestation but also preconception diet studied in infant immune system programming. There are cohort studies describing the influence of pre-conception intake of certain food groups, such as meat- or plant-based food and, except the positive impact mentioned above, it has been stated that pre-conception increased meat intake may increase the risk of wheezing, allergic rhinitis and atopic dermatitis in children [14]. However, it has been also mentioned that the limitation might have been the use of self-reported food frequency questionnaires (FFQs) to assess food intake, possibly leading to errors caused by subject memory bias. Another study, dedicated to maternal pre-conception food intake, such as low and high fat dairy, fresh fruit, saturated and unsaturated spreads, and take-away foods, revealed some protective impact for several atopic diseases but at the same time poultry and fruit juice were adversely associated with eczema, current wheeze and rhinitis classified as “each allergy” [19]. That study was also limited by FFQs and maternal group characteristic showing that almost 30% were obese and 50% were asthmatic. Nevertheless, those studies allowed us to observe some trends: firstly, in the studied issue of the effect of various dietary components for allergy or tolerance development and, secondly, the current way of conducting such research. Most of the studies were focused on the role of a single product or ingredient in pre-conception or gestational diet, not the dietetic pattern driven by health dysfunction appearing during pregnancy. The most common gestational dysfunctions are gestational diabetes and cholestasis. Both usually disappear after childbirth but during pregnancy requires special diet. Also, the gestational vegan/vegetarian diet was tested here. Those diets were declared to be implemented because of the parental high body mass or hyperglycaemia. Finally, the gestational elimination diet was studied in this manuscript. This diet was declared by mothers with their positive history of allergy/intolerance. It was considered to be preventive and assumed an elimination of strong allergens such as milk, wheat, and nuts. In general, there is a limited number of studies evaluating the influence of gestational special diets, such as diabetic, cholestatic or elimination diets, on the specific E antibodies profile in serum of offspring. The vast majority of studies are based on surveys and self-reported manifestations of allergy and other atopic diseases, like asthma, atopic dermatitis or rhinitis, in children [20]. Clinical confirmation of the presence of antibodies and immunological parameters is especially necessary, because some people may be asymptomatic while others will suffer, e.g., as a result of a somatic reaction without immune changes.

The aim of this study was to search for the associations between offspring’s immunological markers like specific IgE profile and cytokines content and maternal intake of special diets, such as diabetic, cholestatic, elimination or based on plants as well as probiotic supplementation. This association was tested both in allergic patients and healthy individuals. This is an important approach due to the increasing prevalence of maternal gestational diabetes, cholestasis, maternal chronic atopic or autoimmune diseases that require following special medical diets. Also, maternal gestational vegan/vegetarian diets applied for medical reasons were tested in this study in the context of offspring developing immune system.

## 2. Materials and Methods

### 2.1. Subjects

The subjects of the study originated from the group of people recruited to the National Science Centre Poland project (No. N312 311939) between the years 2010 and 2014. The project was entitled: “The Influence of Fermented Cow’s Milk Products Displaying Reduced Antigenicity on the Immunological Response of Warmia and Mazury Region’s Patients with Food Allergy with Consideration of Genetic Aspects”. All participants (mean inclusion age 5.9, SD 3.2 years; 56.4% female) were recruited to the project by collaborating medical doctors who were responsible for the food allergy clinical confirmation. All the participants were previously patients of collaborating allergy clinics and the confirmation of allergy was done in in a hospital setting, including provocation tests. 153 allergy sufferers and 150 healthy individuals together with mothers (114 and 108, respectively) were included. Inclusion criteria for food allergy patients were in accordance with the European Society for Pediatric Gastroenterology, Hepatology and Nutrition–ESPGHAN 2015. Briefly: (1) manifestations specific for food allergic disease; (2) confirmed status by immunological tests with increased levels of total and specific serum IgE and cytokines (IL-2, IL-4, IL-8, IFN-γ); (3) positive results of percutaneous skin tests; (4) known allergy family history. Exclusion criteria were: (1) long-term, successful therapy of allergic diseases including desensitization before the recruitment stage; (2) positive history of cancer or inflammatory bowel disease of participants in anamnesis; (3) no possibility to reconstruct the course of pregnancy (no medical documentation); (4) confirmed neurological, psychiatric or systemic illness that might impact cognition or might interfere with retrospective studies. All procedures were approved by the Local Ethical Committee in Olsztyn (Poland)—(decision No. 2/2010) and followed in accordance with the standards of the Helsinki Declaration. Written informed consent was obtained from all parents or statutory guardians of subjects.

### 2.2. Immunological Analyses

Immunological analyses and assessments were performed according ESPGHAN 2015 norms, providing the gold standard in the diagnosis of allergies. All protocols have already been described in the authors’ publications [8]. Briefly: blood and serum samples were collected twice during the project period. Once during the recruitment phase and again after 5 years. All the immunological analyses were carried out using tests certified for clinical trials with declared norms for human samples and certified laboratories. The total immunoglobulin E (tIgE) level of the patients was measured using ImmunoCAP (Phadia AB, Uppsala, Sweden). Food-specific immunoglobulins E (sIgE) were determined using the EUROLINE test with Paediatric and AtopyScreen panels of antigens for allergy diagnosis (EUROIMMUN AG, Lübeck, Germany). Food allergy was confirmed if concentration of sIgE specific to food allergens exceeded 0.7 kU/L (the cut off suggested by test manufacturer: allergy stated with manifestation class ≥ 2). The content of specific IgE antibodies from 0.7 kU/L (class 2) to >100 kU/L (class 6) is considered to be a measurable criterion for both antibodies content and the severity of allergies. It was stricter than those suggested by the ImmuniCAP tests producers that are acceptred in the Project entitled ‘The prevalence, cost and basis of food allergy across Europe’ (EuroPrevall) where sIgE > 0.35 kU/L (only class 1—with no manifestation, was not considered to be a proof of allergy) [21]. Positive serum tests were confirmed in cooperating medical units for the purposes of the study using percutaneous skin tests (Allergopharma, Reinbek, Germany). Both tests were measurable, numerical and comparable. Blood parameters (i.e., absolute number of lymphocytes in the blood and albumin content) were determined by the following authorized laboratories: Stanisław Popowski, Regional Specialized Children’s Hospital in Olsztyn and The Nicolaus Copernicus Municipal Polyclinical Hospital in Olsztyn, using a Cobas analyzer equipped with Cobas MIRA Plus (Roche Diagnostics, Switzerland). Total immunoglobulin G level and serum IL-2, IL-4 and IFN-γ cytokine content were determined using the enzyme linked immunosorbent assay (ELISA) method with commercially available kits (BD Bioscience, USA). Assays ranged from 7.8–500 pg/mL for IL- 2 and IL-4, 4.7–300 pg/mL for IFN-γ and 0.021–15 ng/mL for IgG.

### 2.3. Assessments

Data were collected by collaborating allergologists anamnesis. Medical history complemented with demographics data were submitted by parents/caregivers and by the patients themselves, both in cooperation with the allergologist. The standardized questionnaire for the allergic study corresponding to the validated EuroPrevall study questionnaire included questions regarding family and personal history of atopy, nutrition during pregnancy based on dietary medical indications, supplementation with vitamins and/or minerals, birth method, infant feeding type, probiotic supplementation, current dietary habits, detailed information of the severity of manifestation induced by food intake, applied therapies of allergy, parental smoking and education, diagnosed additional diseases and psychological symptoms (Appendix A) and semi-quantitative frequency questionnaires (Appendix A). It included 133 food items with specified serving sizes that described using natural portions in standard weight and volume measures of the servings. For each food item, participants indicated their average frequency of consumption over the 9 months of pregnancy in terms of the specified serving size by checking one of nine frequency categories. The queries were extended in more detail with regard to cow’s milk allergy area and translated into Polish. Each participant and/or statutory guardian of subject (parents) in examination was provided with information about the processing of personal data and acceptance of the rules (repeated contact for re-assembly of biological material and surveys). The data were collected, processed only for the purposes of the project and in encoded version used in publications.

### 2.4. Artificial Self-Organizing Neural Network Analysis

Databionic ESOM Tools [22] was used to visualize the data set. It is a tool that allows projecting a multidimensional space of input data onto a two-dimensional space of neurons (output data) in a way that preserves the proximity of points. The obtained network can be presented in the form of a U-matrix, giving a graphical representation in which the distances of points from the input space are represented by the colourful terrain map applied in the output space. An image can be interpreted as a three-dimensional “terrain map”, where “valleys” gather adjacent points most similar in terms of analysed features, while “mountain chains” separate the most diverse points (cases). The clustering threshold was designated for <7 cases or according neurones. Using this technique, it was possible to cluster analysis by observing the location of the population of individuals relative to each other. All the measurements were made according to the user manual. All the data were subjected to pre-processing. For each feature the distribution with histograms was done, and if the distribution was skewed (all parameters of specific IgE to inhalation and contact allergens except: mugwort and horse dander, and most of the parameters of specific IgE for food allergens except to eggs) then log-transformation was used. Also, to obtain a uniform scaling of all immunological markers in serum the concentrations data were transformed into percentages. Also, the correlation among features was tested to avoid unwanted emphasis of this aspect of the data. Such features like latex, one of the moulds (*Aletrnaria alternata*) and seafood was discarded from the analysis. In addition, z-transform in Databionic ESOM Tools was used. Pre-processed data were added in lrn format and uploaded to the ESOM Analyser and the default settings of the machine according the training process was used. Briefly, the online training was used which helps to find the best match for a data vector and the neighbourhood on the map is updated immediately. The hybrid threshold and k for kbatch, which is a percentage of all data vectors that is processed between map updates, was set at the value 7. That value describes the minimum percentage of data points that is required to be placed on a different best match, otherwise one epoch of online learning was performed. The number of rows, columns and epochs were respectively 50–80–20, so the number of neurons exceeded 1 K neurons. The search method for best matches was a standard by search. The “quick learning” with constant adding of the current radius of the learning neighbourhood was used. The Manhattan method of weight initialization was applied. Planar topology with borders were prepared. The starting value for learning rate was 0.5 and final was 0.1. The background visualizations with U-matrix were used with best matches (squares). The clustering data both by points and neurons were done manually by grouping points into several clusters through using the ESOM toll. The correlation distances were also tested. The U-Matrix is a visualization of the local distance structure in the data space. Its values are large in areas where no or few data points reside, creating mountain ranges for cluster boundaries. Small values were observed in areas where the data space distances of neurons were small, thus clusters werer depicted as valleys.

### 2.5. Statistics

Absolute frequencies and percentages for qualitative variables and means with standard deviation (SD) for quantitative variables were calculated. The normality of the quantitative variables was evaluated using the Shapiro–Wilk test. For quantitative immunological variables, parametric Student’s *t*-test for comparison among two groups was used, and for more than two different groups the parametric analysis of variance (ANOVA) for normally distributed data and the Tukey test as a parametric post-hoc test or the non-parametric Kruskal–Wallis test in the case of the absence of homoscedasticity were used. For demographic and descriptive parameters of subjects, the comparisons of nominal variables among groups of subjects were made. All tests were two sided, and a *p*-value less than α = 0.05 was considered statistically significant. Also, the Bonferroni’s corrected α = 0.008 value was computed for the number of test *m* = 6 calculated for 4 compared groups.

Correlation between quantitative variables was evaluated using the Pearson’s correlation coefficient (r) and that between nominal variables was evaluated using the Spearman’s rank correlation coefficient.

To determine the relationship between the analysed data and to indicate the major factors that influence the profile of determined allergy, principal component analysis for quantitative variables and multiple corresponding analysis for standardized qualitative variables was carried out [23,24]. The allergy type and immunological criteria were used as the grouping variables. Immunological parameters results (cytokines content, lymphocytes subpopulation, and serum albumin content) were applied as quantitative supplementary variables for dietary/specific IgE associations. When choosing a number of significant components, two criteria were taken into consideration: the Kaiser criterion and the number of major factors explaining at least 90% of total variability.

Both univariate and multivariate analyses were performed using the statistical package STATISTICA 13 software (StatSoft, Cracow, Poland) and XLSTAT for Excel software (Microsoft, Albuquerque, NM, USA).

## 3. Results

### 3.1. Sociodemographic Associations

Of the 153 allergic patients enrolled in this study, subjects suffering from allergy to a single food-origin allergen (SFA) and polyallergy to various food-origin allergens (PFAs) were 31% each (Figure 1). Mixed polyallergy to aero- and food allergens (MFAs) was determined in 38% of sufferers. The gestational diet, habits and health condition of 114 mothers of allergic patients and 108 of healthy individuals were analysed. The age–sex structure between the allergic and control group of participants did not differ significantly, even though the study was not randomized (Table 1). In each allergic group and in the control, over 50% of participants were female and the mean age of study entry was 5.1–6.7 years for allergic and 6.8 years for control. More frequent PFA occurrence in the female group (*p* = 0.007) was observed. No further associations of age with variables were noted. The place of residence had no significant effect on the allergy occurrence but the tendency for more frequent diagnosis of SFA in urban places was observed. Simultaneously, the moderate positive correlation (*r* = 0.503, *p* < 0.005) between living in the city and gestational dieting and with BMI abnormalities (*r* = 0.313, *p* < 0.005) in polyallergic groups (PFAs, MFAs) were stated. Slight negative correlations among those variables in the SFA group was noted (*r* = −0.374; *p* < 0.005).

### 3.2. Health Status

An allergy prevalence in subjects with a positive family history of atopy was significantly higher (*p* < 0.001) than in negative subjects (Table 1). Positive family history of atopy was moderately positively correlated with the frequency of gestational elimination diet (the exclusion of ingredient followed by mother only in pregnancy, most often reported was milk, wheat or nuts elimination, caused by maternal allergy, or positive family history of allergies) also in the control group (*r* = 0.596; *p* < 0.005). A weak but significant correlation was also observed between positive maternal chronic diseases history and diagnosing chronic diseases in children (*r* = 0.377; *p* < 0.005).

Body Mass Index (BMI) disproportions of subjects with allergy were significantly more common than in control (*p* = 0.006). The disparities were especially visible in the MFA group, where underweight cases reached 35% and overweight cases reached 40%. A positive association between the frequency of mixed infant feeding type, several food allergens elimination diet of subjects and the frequency of their BMI abnormalities like underweight, overweight (*r* = 0.476; *p* < 0.005) were observed.

The severity of allergy manifestation was moderately correlated with the tendency towards maternal diets’ neglect of adherence (*r* = 0.489; *p* = 0.005). Nevertheless, it has shown no dependence of severity of allergy symptoms and the type of allergy (Table 1) and subjects’ serum IgE level. The severity appeared to be strictly individual tendencies.

### 3.3. Gestational Diets and Supplementation

Only 10–32% of pregnant women whose children were included in allergic groups consumed a balanced traditional diet, and in the control group that percentage reached 49% (Table 1; Appendix A). Almost 25% of women declared following even two gestational dysfunction-driven diets (cholestatic with diabetic; or elimination with vegetarian or with diabetic). This was common for women suffering from pregnancy hyperglycaemia and cholestasis at the same time or women on an elimination diet at the same time with pregnancy hyperglycaemia.

Depending on the group of subjects, the gestational elimination diet followed from 10 to 38.3% of mothers but also, prophylactically about 6.7% healthy women whose children did not develop allergies, but the family history was enlightening (Table 1; Appendix A). Maternal gestational diabetes diet intake was declared in over 30% of the tested population and in the PFA group the ratio was the highest. Moreover, in that group it commonly co-occurred with cholestasis. No relationship was found between the age of mothers and the frequency of these diets.

The second type of gestational diets were habitual diets (like vegan/vegetarian) -the use of which during pregnancy was not only due to medical reasons. Those diets were commonly used with the intention of increasing immunity and instead of or together with elimination (milk, wheat and nuts) diets and were declared by 2–13% of women (Table 1; Appendix A).

There were no significant differences in the frequency of vitamins and/or minerals supplementation but over half of women confirmed the use of supplementation during pregnancy. The most often, supplementation with folic acid, biotin and vitamin D_3_ as well as magnesium and calcium was indicated by women whose children were diagnosed as PFA. This regularity of increased supplementation was observed in the group most often following gestational diets.

Significant differences in the frequency of gestational supplementation with probiotics were noted in the groups. Over 35% mothers of children with stated different types of polyallergy confirmed using commercially available probiotics. That intake was medically consulted. The methods of intake as well as bacterial strains composition were varied but the use of medically justified, vaginal probiotics was confirmed relatively frequently (about 48% of all cases). Those probiotics were dedicated to use for women with frequent gestational genital tract infections.

### 3.4. Birth, Infancy and Subject Diet

Over 50% of the children taking part in the studies were born naturally, but the lowest percentage was noted in the MFA group. The differences among groups were not significant. Visible discrepancies were stated in the declared feeding type of the newborn. Exclusive breastfeeding was used two times more often in the SFA and PFA groups than in the MFA; however, there was no significant difference between this group and control. Mixed feeding along with Caesarean section was reported most often in the MFA group. In that group of children, over 33% was supplemented with commercially available paediatric probiotic.

It was also observed that in the MFA group, an elimination diet was the least frequently used in children; however, if the elimination was already implemented, then in more than 70% of cases, it included several food allergens.

### 3.5. Immunological Parameters

Significant differences in total IgE concentration not only between particular allergic groups and the control (*p* = 0.017) but also regarding the sex of the patients (*r* = 0.412; *p* = 0.005) and total IgG (r = 0.393; *p* = 0.005) (Figure 2a) were observed. The ratio of total IgE/total IgG in the tested group of allergic subjects discriminated two clusters. 59.47% of cases had high total IgE but lower total IgG content. The second variable, describing 38.57% of cases, was determined by moderate but increased age–norm of total IgE and high total IgG content.

Higher levels of IgE were observed in male groups (mean total IgE–728.2 ± 98.3 kU/L and 20% of male participants showed a reaction in class 5th of specific IgE and higher, while in the female group it was only approximately 4% of cases and the mean total IgE was 412.3 ± 56.3 kU/L). The differences in male group were dependent on the type of specific IgE; for male, it was mainly grass, birch pollen, mites (Appendix A). In female subjects, such associations were not observed. Female MFA subjects could demonstrate moderate serum total IgE amount, but in 4–6 classes of specific antibodies it was only 25% of cases. Both antibodies’ profiles were positively correlated with cytokines’ profiles: IL-4 and IL-8 (Figure 2b) and IgG/IgE with IL-2, IFNγ (Figure 2c). Moderate correlation was also observed between content of IFNγ and IL-4 (r = 0.496; *p* < 0.001), and IL-8 (*r* = 0.365; *p* < 0.001) (Appendix A).

The secretion of analysed cytokines and antibodies was elevated despite 30% of the allergic patients having been diagnosed as suffering from lymphopenia and only 8% had lymphocytosis. Both abnormalities in lymphocyte levels were significantly higher than in control group (*p* = 0.031 and *p* = 0.048, respectively). The reduced absolute number of lymphocytes in the blood and lower concentration of albumin (in 15% of allergic patients) were most frequently observed in subjects who were following a multiple product elimination diet. Observed differences in levels of both blood parameters showed interdependence with subjects’ BMI but also total immunoglobulin level and it was a moderately negative correlation (*r* = –0.473; *p* = 0.005).

In the serum of the majority of SFA patients, specific E antibodies against nuts (f13, f17), soybean (f14), cows’ milk (f2) and, especially, casein (f78) were dominant and reached 3–6 classes, but the most often diagnosed food response was caused by potato (f35) (antibodies in 2–4 classes) (Figure 1 and Figure 3). Importantly, almost 60% of those subjects were on an elimination diet of a single allergen but 15% were on several allergens elimination diet prophylactically implemented by parents on their own. In the PFA group, proteins of egg white (f1) and yolk (f75), codfish (f3), wheat flour (f4), soybean (f14) and potato (f35) were the most common allergic irritants. Egg white, codfish and soybean gave even 5th and 6th class responses. Several food allergens’ elimination diet followed almost 40% of allergy sufferers when 25% declared elimination sources of single allergen—the vast majority pointed to eggs. In the MFA group, over 60% of subjects showed increased specific IgE antibodies content to birch pollen (t3), dander of cat (e1), dust mites: Dermatophagoides (d1 and d2) and moulds (m3 and m6) along with food origin allergens to eggs, milk and nuts. In that group, the elimination diet was also the least restrictive.

In the studied population, the most common cross reaction was birch–potato (42%). The rest noted were pollen–nuts (38%); mugwort–carrot–apple (32%), dust mite–seafood–fish (15%), milk– meat (15%) and latex–citrus (7%). No direct effect of positive atopy family history and birth method on total subjects’ IgE was observed; however, there was a weak positive correlation between newborn feeding type and total IgE content in all subjects including control individuals (*r* = 0.319; *p* = 0.005).

The total IgE level of subjects did not show any correlation with maternal antibody level, while the level of total IgG showed weak positive correlation (*r* = 0.311; *p* = 0.005). This regularity was also observed between the serum IgG levels of mothers and offspring in the control group. The level of IgG in the sera of cases in control group was not significantly increased and it was not associated with any additional, referred chronic disease. 

### 3.6. Gestational Dysfunction-driven Diets Associations

A major adjustment chosen, based on previous immunological clustering, was high/low contents of immunoglobulins E and G (Figure 2a). In this adjustment, high IgG/moderate IgE emerged as a grouping variable. Other immunological parameters have been treated as an additional variable. As a result of the PCA analysis of the data, it was found that the first two components explained 38.57% of variations (Figure 3a) and to describe 90% variability, nine factors were needed. Multivariate analysis showed a high inter-individual variation of the data, but in the first adjustment two main factors (Figure 1 and Figure 2) grouped population on seven coherent clusters marked as i–vii (Figure 3b). The coherence of clusters was tested using Dunn test to estimate the average distance between clusters. The significance of associations in clusters was tested using analysis of correlation the results of which are presented in the Appendix A. The contribution of variables in major factors were demonstrated in Figure 3c. The contribution of other factors is presented in Appendix A.

For 29.59% of changes, a newborn probiotic supplementation and gestational diabetic and cholestatic diet of mothers were responsible. In that factor, the dominant contribution of specific IgE profile had antibodies characteristic for SFA and PFA like anti-potato, -egg, -nuts, and -soybean. The high contribution of anti-pollen IgE was also noted in MFA clusters. For 8.98% of variability, meat was associated with a newborn’s formula feeding type and cross-reactions like latex–citrus, dust mites–fish–seafood but also cow’s milk proteins.

Clusters separated in the form of an artificial neural map (Figure 3d) grouped subjects in 7 clusters. An exact description of clusters is contained in Appendix A). There was negative correlation between total IgE and probiotic supplementation of subjects (*r* = −0.411; *p* = 0.005). Gestational diabetic and cholestatic diets were more differentiating in the “moderate IgE but high IgG” group of subjects. Subjects with class 4–6 of specific antibodies to eggs and nuts were more often found in groups of children born by mothers that followed gestational cholestatic diets (clusters c and d). Moderate to high (class 2–5) levels of antibodies to apples, carrot, potato, wheat flour, and pollen were determined in groups of children whose mothers were on gestational diabetic diet (clusters a, c and e). High levels (class 5–6) of antibodies to nuts and soybean were more common in children whose mothers follow vegan/vegetarian diets (clusters d and f). Moderate-to-severe reactions to milk (class 2–5) were observed in children of mothers who followed gestational elimination and diabetic diets (clusters a, d and g). Overweight BMI status was associated with cross-reactions and infant formula based infant feeding in cluster b, whereas the normal and underweight BMI status was associated with cross reactions and breastfeeding in cluster e. In the entire analysis the cluster coherence exceeded 80%

### 3.7. Gestational Elimination and Vegan/ Vegetarian Diets Associations

In this adjustment, “high IgE, moderate IgG” emerged as a grouping variable (based again on immunological clustering Figure 2a), and the rest of immunological parameters were treated as an additional variable as in Section 3.6. The first two components explained 46.67% of variations (Figure 4a), and to describe 90% variability, seven factors were needed. Multivariate analysis showed a high inter-individual variation of the data and PCA analysis grouped population only on four overlapping clusters marked as I–IV (Figure 4b), whereas the artificial neural network allowed observation of 19 coherent clusters (Figure 4d). The contribution of variables in major factors (Figure 1 and Figure 2) were demonstrated in Figure 4c. The contribution of other factors is presented in Appendix A. The detailed description of clusters and the significance of associations in clusters was also tested using analysis of correlation, the results of which are presented in the Appendix A.

For 30.46% of changes, a maternal probiotic supplementation and elimination diet, but also habitual gestational vegan/vegetarian diets of mothers, were responsible. In Figure 1, the dominant contribution of specific antibody had characteristics for MFA like anti-pollens, -mites and -molds but also for PFA anti-nuts, -soybean, -potatoes, -eggs and -milk. The high contribution of a subject’s gender was also noted. For 16.21% of variability, a newborn’s formula feeding type and subject’s elimination diet and also maternal gestational diabetic diet were responsible. The high contribution of subject’s BMI was also noted. In Figure 2, the dominant contribution of specific antibody profiles were those characteristic for MFA like IgE against pollens and dander but also for SFA – milk, beef, tomato. There were positive associations between gestational probiotic supplementation of mothers and specific IgE content of subjects (*r* = 0.498; *p* = 0.005) and infant formula feeding type (*r* = 0.462; *p* = 0.005). Gestational habitual and elimination diets were more differentiating in the high total IgE group of subjects. Clusters separated in the form of an artificial neural map (Figure 4d) grouped subjects in 19 clusters. Exact description of clusters is contained in Appendix A.

Subjects with class 1–4 of specific antibodies to rice, class 2–5 to nuts and soybean, and class 5–6 to dander were more often found in groups of children born by mothers that followed elimination diets (clusters C, E, G and S). Moderate to high (class 2–5) levels of antibodies to apples, carrot, tomato and potato, and high (class 5–6) to pollen were determined in groups of children whose mothers were on gestational diabetic diet (clusters D and J). Moderate to high (class 3–6) level of antibodies to nuts, soybean, eggs and moulds were more common in children whose mothers follow vegan/vegetarian diets (clusters K, L and *p*). Subjects with class 5–6 of specific antibodies to nuts were more often found in groups of children born by mothers that followed gestational cholestatic diets (clusters M, O and Q). Overweight BMI status was associated with pharmacological treatment (clusters E and *n*), whereas underweight BMI status associated with multiproduct elimination diet in clusters D, G and J. Female sex seemed to have an impact on severe reaction to animal dander and pollen, and high level of antibodies to soybean and milk (clusters C and J), whereas male sex groups demonstrated high level antibodies (class 5–6) to grass, mites and tomatoes (clusters D and K). Infant formula feeding associated with high total IgE was observed in 6 clusters (A, H, J, M, P and Q). Maternal probiotic supplementation associated with high total IgE was observed in 10 clusters (A, C, D, E, L, M, O, P, Q and S). In the entire analysis the cluster coherence exceeded 80%.

## 4. Discussion

Our study, for the first time, reports the profile of specific IgE antibodies, total concentration of several immunological components such as total immunoglobulins E and G, and proinflammatory allergic cytokines in both allergic and control groups of subjects in the context of gestational habitual and dysfunction-driven dietetic patterns in observational studies.

A total of 54 variables describing the content of specific IgE antibodies and eight immunological parameters with 20 demographic and descriptive parameters were integrated to better understand the effect of gestational factors on the development of an allergy profile or tolerance in offspring; these parameters included subject’s age, gender, occupation, BMI status, infant feeding type, maternal education, income level, health status, personal history of atopy, nutrition during pregnancy, diet supplementation, birth method, family atopy history, and co-occurring chronic diseases including psychological symptoms, smoking, applied therapies of allergy, and severity of atopic symptoms. Some of these variables were qualified by different groups using multidimensional analyses of major components [25], but a much more efficient tool for data mining seemed to be ESOM-Maps tools for clustering, visualization, and classification of complex data. In most studies reported to date, an individual’s food intake and different food patterns emerged to be associated with allergic outcomes; however, this was mostly based on self-reported allergic diseases and skin prick tests, with no clinically determined level of specific antibodies [26,27]. In our present study, the number of variables was much higher. The most common method for complex data projection in this area is the t-distributed stochastic neighbour embedding (t-SNE), but this approach might be frequently unreliable in the context of biomedical, cytometry data analysis because of the incorrect number of subgroups or projected data points belonging to different subgroups, as if they belong to the same subgroup. Novel computational techniques allow to visualize structures in these data and to eventually identify relevant subgroups. It was postulated to use emergent self-organizing maps in combination with U-matrix methods, which is much more precise and appropriate for complex biomedical data [28]. This alternative approach in our studies also seemed to be more efficient. The original data sets shown in the principal component analysis also allows to determine the main trends in such a large subset of data and variables. However, to meet the actual EAACI guidelines on monitoring the diet diversity in pregnancy, infancy, and childhood in the progression of atopic diseases, a more efficient and precise tool seems to be the one appropriate for our applications [20]. The EAACI report highlights that there are limited data on diet diversity in pregnancy and allergy outcomes in the offspring and the majority of studies refer to asthma/wheeze in the offspring [20]. There are also reports on diet quality in infancy, but they have shown a possible association with reduction in wheeze/asthma severity following consumption of single products and rarely with diet diversity [29]. In contrast to our studies, all those studies examined different characteristics of health status and diet diversity and used different instruments for data collection and analysis.

### 4.1. Gestational Dietetic Patterns and Allergy Associations

Our research was focused on nutritional patterns, not on particular nutrients and types of food. Basically, nutrients and foods are not eaten in isolation, and excessive intake of one of them may inadvertently lead to reduced intake of another and to cause dietary imbalance. The most commonly used nutritional patterns in the determination of allergic disease associations are those that are geographically based, such as Mediterranean, Oriental, Japanese and Western patterns [30]. Additional pattern classification is focused on the degree of food processing and health criterion: traditional, processed, confectionary, healthy, and health confident [27]; and the last type of classification of atopic-associated patterns is based on the dominant food ingredients: high carbohydrate, protein, plant-based and ketogenic patterns [31]. All these patterns are, however, not driven by gestational dysfunctions, and despite common nutritional elements, they still do not focus directly on the gestational medical reasons for their use.

Few studies have considered the impact of offspring, not maternal, diabetic diet on atopy prevalence in children [29]. A significantly increased incidence of atopy was explained by lesser diversity of diet. Nwaru [29] also emphasized that the study was focused on the group of subjects with type I diabetes in a Finnish population. Children recruited in that study carried genetic HLA-conferred susceptibility for type 1 diabetes, and they emerged from the cohort. In these studies, HLA-DQB1 was considered to be a genetic susceptibility locus not only for type 1 diabetes mellitus but also for allergic sensitization which might have been the reason for higher correlation of all the tested variables. Another tested type of health-driven diet was allergens’ elimination. Several studies have investigated the impact of elimination diet on the frequency of allergy in offspring; most of these studies have not supported the protective effect of maternal exclusion (including the exclusion of cow’s milk, eggs and peanuts), but some of them have reported mild preventive action on cross-reactions [32,33]. The impact of low-fat cholestasis diet was not tested in the context of allergy and other atopic disease association; however, some studies have reported the protective effect of gestational low-fat dairy consumption on the risk of wheeze. These results, however, seem to be inconclusive because of the inclusion of other immunomodulatory components (peptides, bacterial cells, SCFA) in those dairy products [34]. Most studies have emphasized the protective effect of high-fat products in the development of asthma and atopy, but it was also based on the type and composition of specific fatty acids. Dominant characteristics were long-chain polyunsaturated fatty acids from fish and some plant-based oils such as olive oil. Vegetable oils and margarine were reported to be products associated with a higher risk of allergic diseases [35]. In our research, because of the lack of similar reports, we attempted to analyse the declared diet composition associations based on previously collected patterns and individual diet components.

In the group of mothers remaining on a diabetic diet, the dominant consumed products were in the so-called healthy pattern, and dominant products were vegetables, whole grain bread, fermented dairy products, eggs, lean fish, and minimally processed foods/meats such as poultry. The limited consumption of potatoes, some root vegetables such as carrots and beets, white bread, rice, raw milk, and fruits in that pattern caused the high glycaemic index not indicated in hyperglycaemia. This nutritional pattern in our studied group of subjects was associated with an increased level of specific E antibodies to potato, carrot, apple, cow’s milk, and eggs and also to birch pollen and dust mites due to cross-reactivity.

Excluding some components of the diet and increasing the consumption of others might influence the IgE profile of offspring. Earlier protective influence of maternal and subjects’ high fruit consumption and reduced consumption of pasta, fast food, and potatoes on wheezing, dermatitis, and rhinitis has been reported. The prevalence of disease was 18.8%, 17.2%, and 10.4%, respectively, and in each group, a higher risk of diseases was observed for low fruit consumers and high carbohydrate consumers [36]. Maternal high intake of green and yellow vegetables, citrus fruits, and β-carotene was also reported to be associated with a lower risk of eczema [26]. The levels of immunological parameters were, however, not reported, and no direct association with allergy was studied. In our studies, the increased presence of specific antibodies to maternal excluded products was moderately correlated (*r* = 0.412; *p* = 0.005), and other observations included a high level of IL-2/IL-4 in a group with increased total IgE (r = 0.439; *p* ≤ 0.001), and IL-8/IFN-γ (*r* = 0.5; *p* ≤ 0.001), in high total IgG group Appendix A. The majority of cases in that group declared a negative family history of allergy. This immunological profile of subjects can be caused by a reduced consumption of natural sources of not only cofactors and methyl donors, such as vitamin B6, B12, folic acid, and betaine, but also omega-3 fatty acids that could alter the methylation of de novo genes in utero. A previous study reported that maternal nutrition and the intake of these components play a critical role in DNA methylation in utero and the development of the offspring’s immune system in the course of many diseases [37]. In dual transcriptomic and epigenomic study by Ba et al. [37] in which 318 genes with changes in their expression and only CD4+ lymphocytes were considered, 203 CpG sites with differential DNA methylation were found to be associated with the process of peanut allergy development and regulation. The results of previous studies were based on the type of allergy, the involved tissue, and the severity of manifestation [38]. These studies stated that methylation may serve as an anchor upon which gene expression modulates reaction severity, but depending on the gene, both increased and reduced methylation can be a factor in the induction of an allergic process and its severity. NF-κB and its inhibitor alpha (*NFKBIA*) seem to play an important role here. *NFKBIA* and *ARG1* act as hubs in the networks, and 3 groups of interacting key node CpGs and peanut severity genes are involved in immune response, chemotaxis, and regulation of macroautophagy [38]. Another study showed that transient exposure to hyperglycaemia induced long-lasting activating epigenetic changes in the promoter of the NF-κB subunit p65 in aortic endothelial cells, both in vitro and in non-diabetic mice. An increased frequency of monomethylation of histone 3 lysine 4 caused an upregulation of p65 gene expression, leading to a sustained increase in the expression of the NF-κB-responsive genes *MCP-1* and *VCAM-1* [39].

In the group of mothers remaining on an elimination diet, because of their positive history of atopy and current allergies and intolerances, the dominant consumed products were in the traditional pattern and the dominant products were vegetables, processed red meat and poultry, potatoes, and white vegetables. This group declared exclusion of milk, eggs, flour products, and nuts. This nutritional pattern in our studied group of subjects was associated with increased level of specific IgE antibodies to cow’s milk, potatoes, and root vegetables and to pollen, dust mites, nuts and dander; however, the total IgG class antibodies was in the normal range. Our results are comparable to those observed in another large population-based study of women of fertile age in southern England and in ALSPAC reports. A similar association between the traditional diet and allergic diseases was observed in the previous study with declared allergic symptoms to products such as root vegetables and potatoes (*r* = 0.321–0.704) [27], while in our studies, we confirmed such association in terms of specific antibodies to those allergens (*r* = 0.411–0.515). Our results also confirmed previous reports which indicated no significant preventive effect of maternal allergen avoidance on offspring and it seems to have an opposite association that such maternal behaviour worsened the condition of the offspring [40]. In cited studies, higher maternal peanut intake during the first trimester was associated with 47% reduced prevalence of peanut allergic reaction. Higher wheat intake during the second trimester was associated with reduced atopic dermatitis, whereas higher milk intake during the first trimester was associated not only with reduced asthma and allergic rhinitis but also with reduced cow’s milk allergy of offspring [41]. All the above mentioned results suggest the protective influence of maternal allergen-specific IgG antibodies on offspring which was also observed in the context of gestational allergen-specific immunotherapy (AIT) and placental transfer of allergen-specific IgG1 and IgG4 antibodies, but not IgE, from AIT-treated mothers to their offspring. For 164 tested allergens, children from mothers with increased (>30 ISAC standardized units) specific plasma IgG levels against an allergen developed no IgE sensitization against that allergen at 5 years of age [42].

In the group of mothers remaining on cholestatic diet, the dominant consumed products were in the health-conscious pattern, and the dominant products were green vegetable salads, fruits, juices, dark rice, pasta, oat/bran, mushrooms, and minimally processed foods/meats such as poultry and lean fish. The limited consumption of products with high fat content such as red meat, eggs, nuts, cheese, butter and vegetable oils and products with high glycaemic index was declared. In the serum of offspring, specific IgE antibodies to eggs, cows’ milk protein (especially casein and bovine serum albumin (BSA)), and nuts were frequently observed. In the group of mothers on double gestational diets, specific antibodies to fish, seafood and beef proteins were found, but these were additionally correlated with newborn infant formula feeding. Moderate-to-high levels of IgG, IL-2 and IFN-γ were observed in that group of offspring. It has been previously reported that diets rich in fat and fat-soluble components, such as vitamin E, proteins, hormones, and polyphenols, may alter the expression of regulatory miRNAs. In contrast, restrictive cholestatic diet along with diabetic diet contributed significantly to the limited consumption of these components, but at the same time, it may cause disorders in the body’s lipid metabolism. The deficiency of these components may cause a downregulation of miR-122a and miR-125, which contribute to altered regulation of lipid metabolism and inflammation, respectively [43]. miR-125a plays role in NF-κB regulation, while miR-125b regulates the proliferation and activity of B lymphocytes. It also plays a significant role in inflammatory state and skin remodelling, especially in psoriasis and atopic dermatitis and in eosinophilic chronic rhino sinusitis through IFN overproduction [44]. Similar to our results, some reports showed that the maternal health-conscious pattern is related to an increased risk of early and late eczema, atopy, and increased total IgE level [27]. Similar interactions have been observed with declared allergic symptoms to products such as flour products (both white and full grain), rice, fruits, fish, eggs and cheese (*r* = 0.443–0.615), while in our studies, such associations were observed in terms of specific antibodies to allergens (*r* = 0.320–0.589).

In our studies, following a gestational vegan/vegetarian diet was considered as habitual, but it was also partly motivated by mothers’ health condition and alleviation of symptoms of diseases, i.e., allergy, intolerance, and autoimmune diseases including diabetes; this finding is consistent with studies of other groups [45]. In the vegetarian/vegan pattern, the dominant products were vegetables, fruits, juices, dark rice, and meat substitutes based on pulses, nuts, grains and herbs. Consistent elimination of animal products from the diet was declared. This seemed to be associated with high levels of total and specific antibodies of class E to soybean, nuts, beef, eggs, cows’ milk proteins and pollen. In these subjects, high levels of interleukins were also found, even in patients not manifesting allergies from the control group. Some reports suggest that early exposure to phytoestrogens can increase IgE production later. Our observations were consistent with ALSPAC reports and the scale of the observed phenomenon was at a similar level (*p* = 0.029 and *p* = 0.026, respectively) [27]. Previous studies by Guo et al. [46] testing in utero exposure to genistein in a mice model showed an enhancement of respiratory allergen with an increased IgE production; decrease in the percentages of CD4+ CD25+ T suppressor cells; and increase in the NK cell activity, the basal splenocyte proliferation, the expression of CD86 by B cells, and the production of IL-2 and IL-4 [46]. Additionally, individuals on plant-based diets are at a risk of nutritional deficiencies of modulators such as vitamin D, omega-3 fatty acids, calcium, iron, and vitamin B12; hence, these diets require a strong awareness for a balanced intake of key nutrients [45]. High folate intake may cause vitamin B12 depletion by rectifying the hematologic alterations that are present with vitamin B12 insufficiency. During pregnancy, the intestinal absorption of vitamin B12 increases, but it runs efficiently in small, frequently taken doses. Vitamin B12 derived from maternal tissue reserves does not cross the placenta, in contrast to the absorbed form. Deficiency of B12 may lead to hyperhomocystinemia and to global DNA hypermethylation, as it has been reported in the prevalence of many diseases [47].

In our studies, the association of high total IgE and cytokines profile of offspring and gestational plant-based diet, as noted in other studies, might have also arisen from multiple additional factors and analysed outcomes such as infant feeding type and birth method. As postulated previously, this finding along with the few other associations that achieved nominal significance should be examined in further studies.

### 4.2. Gestational Probiotic Supplementation, Microbiota Modulation, and Allergy Associations

The majority of studies highlight the importance of commensal bacteria in host–immune response modulation and the important role of probiotics in this process. Some in vivo studies explain the mechanisms of gestational programming of offspring through microbiota modulation with relevant role of maternal microbiota composition and antibiotic treatment [48,49,50,51]. These studies were most often conducted on mouse models, but people with chronic allergies were used as microbiota donors. In one of the experiments, pregnant mice fed with high fibre diet showed changes in microbiota composition, increased number of *Bacteroidetes* and high production of short-chain fatty acids. The increase in the latter, especially acetate, in dams during pregnancy induced Treg in offspring mediated by enhancing acetylation of Foxp3 promoter through the inhibition of histone deacetylase 9 in utero; this suggested that maternal microbiota influence the susceptibility of offspring to allergies through epigenetic modification in utero [52]. Maternal factors directly or indirectly affect the immune system of offspring during pregnancy. Recent studies on intestinal dysbiosis are focused on its association with susceptibility to food allergy and its heredity. *Anaerostipes caccae*, a butyrate-producing *Clostridial* species, which originated from allergic children, was reported to be critical for protection against food allergy. The administration of these immunomodulatory bacteria to the ileum of cows’ milk protein-immunized mice caused reduction in antigen-specific, Th2-dependent antibody (serum β-Lg-specific IgE and IgG1) and responses of cytokines IL-13 and IL-4. The abundance of these bacterial cells in ileal samples was also correlated with two most differentially expressed genes, *Acot12* and *Me1*, which are involved in pyruvate metabolism of epithelial cells. In immunized organisms, the downregulation of butyrate production in this strain was also observed. Therefore, these bacteria were considered to play an important role in programming the epithelial barrier physiology and state of allergy and tolerance of the host organism on many levels [53]. In other studies, the role of bacteriotherapy in the activation of MyD88-dependent microbial sensing pathway in nascent Treg cells that upregulate disease-suppressing ROR-γt + Treg cells was postulated [54]. In these studies, *Clostridial* species, impacted by dysbiosis in human infants with food allergy, could suppress food allergy after transfer to immunized mice through various mechanisms. It has been observed that treatment with the *Clostridial* consortium increased sIgA response to the microbiota and suppressed the IgE antibacterial response. The authors postulated that this change was due to the regulatory effect of intestinal mucosa metabolism on the immunoreactive nature and metabolism of bacteria. The disruption of the MyD88-ROR-γt regulatory axis by dysbiosis in food allergy organisms led to decreased IgA and increased IgE responses to the gut microbiota, which were reproduced upon Treg cell-specific deletion of *Rorc*. Abdel-Gadir et al. [54] (2019) suggested that an anti-microbiota Th2 response may be elicited during hypersensitive immune response to foods and may play a critical role in disease initiation, persistence, and outcome [54]. This finding was convergent with our previous studies, indicating the ability of IgE antibodies of patients with cow’s milk protein allergy to bind to proteins of *Lactobacillus*. We confirmed that cyclopropane-fatty-acylphospholipid synthase and carboxylate-amine ligase in *Lactobacillus casei* LcY react with human IgE antibodies [55]. In the context of these findings and the reported presence of bacterial cell components and metabolites in the placenta, it might affect the foetal immune system in utero [56,57]. In our current studies, gestational probiotic supplementation yielded ambiguous results. It did not alleviate the severity of allergies in offspring (the levels of total IgE and cytokines were still high). This could be caused by the varied composition of applied probiotics, dose, reason, and duration of their use, and the route of administration. Some women reported temporary use of gynaecological probiotics because of previous infections, and this variable was strongly positively associated with high total IgE and specific IgE levels to pollen, dust mites, and BSA. A significant protective effect was observed because of subject’s probiotic supplementation. The analysis of intestinal microbiota composition of subjects along with their mothers in this context would be useful.

### 4.3. Other Associations

In the studied population, some variables strengthened the observed associations of allergies and gestational diet and supplementation. One of these variables was sex of subjects. Both the incidence and profile of specific IgE tended to be gender dependent, which has also been observed in other studies [50]. The profile of specific antibodies seems to be dependent on both gender and the geographical origin of the population; however, our studies did not agree with those reported for profiles of specific antibodies in the serum of Indian and Chilean offspring. This phenomenon may be influenced not only by microbiota composition as discussed earlier but also by sex hormone- and metabolic gender-specific factors that may contribute to the differences in food allergy occurrence and manifestation [58].

One of the variables that strengthened the association was BMI status. The observed disorders (obesity and underweight) were previously reported in the group of these subjects and were partly related to co-existing eating disorders because of allergy therapies and following different types of diet [8].

Another factor is the maternal history of nonallergic disease burden. An interesting issue seems to be the intake and metabolism of various dietary components in the pre-conception period and during pregnancy and lactation. One of the recent studies reported that maternal food intake during pre-conception, such as low- and high-fat dairy products, unsaturated spreads, fresh fruits, and take-away foods, were protective for any allergy assessed. Consumption of highly processed meat, high-sugar diet, and legumes was positively associated with allergy. Saturated spreads (e.g., butter) were protective against eczema, current wheeze, and rhinitis, while non-oily fish was protective against eczema and current wheeze; however, poultry and fruit juice were adversely associated with allergies. The authors, however, emphasize that this topic requires further research [19]. Nonetheless, these results coincide with the false alarm hypothesis, which suggests that food allergy is associated with high intake of advanced glycation end-products (AGEs) and proglycating dietary sugars that may work as alarmins—endogenous molecules secreted from cells undergoing nonprogrammed cell death. The Western diet is rich in AGEs, which are consumed with processed meat and oils. AGEs are also formed in the presence of a high concentration of sugars. Their high intake results in misinterpretation of a threat from dietary allergens, through AGEs and other alarmins and induction of innate signalling through multiple mechanisms, resulting in the development of allergic phenotypes. Untreated gestational diabetes might be associated with increased foetal exposure to glycation products formed from greater glycation of endogenous proteins through a nonenzymatic mechanism; this was reported to be associated with an increased risk of atopic dermatitis and IgE sensitization [59]. Few studies have focused on the role of lipid metabolism in regulating the sensitization process and atopy. Mice fed a ketogenic diet showed dietary restriction of glucose, impaired lipid metabolism, and ablated innate lymphoid cells 2 (ILC2s)-mediated allergen-driven airway inflammation. Innate lymphoid cells are involved in the maintenance of barrier immunity, but their chronic activation results in immune-mediated pathology. This metabolic program is imprinted by interleukin-33 (IL-33) and regulated by the *Ppar*γ and *Dgat1* genes, both of which are controlled by glucose availability and mTOR signalling. Reduction of glucose and increased consumption of lipids might prevent chronic activation of ILC2s [60]. Thus, all the mentioned diets have their pros and cons, and following them strictly may trigger various allergic mechanisms. It is important to precisely adapt the diet to the mechanism of the disease. Therefore, appropriate therapy for pregnancy syndromes is necessary as well as the awareness that many factors and mechanisms affect the immune system of the offspring.

### 4.4. Limitations and Directions for Future Research

The study participants were recruited from the north-eastern region of Poland. It was representative of the communities from which it was sampled, but probably not of the entire population. Therefore, findings should be extended and confirmed in studies that include participants from other regions. Nevertheless, the Polish population is perceived to be rather homogeneous in socioeconomic and racial terms. Another limitation might be the use of self-reported semi-quantitative food frequency questionnaires to assess food intake, which could possibly lead to errors caused by subject memory bias. Nevertheless, dysfunction-driven diets are characterised by limited diversity and strict adherence, and potential undesirable consequences of diet derogation may cause severe burden. The regularity and composition of meals are also more controlled. Nonetheless, the present study provides a strong background for future research. Further research on the role of microbiota in the studied population should be conducted, especially in the context of intriguing results related to probiotic supplementation.

## 5. Conclusions

In summary, this study indicated that gestational dysfunction-driven diets and probiotic supplementation determine the profile of allergen-specific antibodies in the serum of allergy sufferers. Significant associations were observed between special gestational diet intake underlying foetal programming and the mechanisms of childhood allergy. The novelty of this research study is the positive association between diabetic and cholestatic diet intake and mixed IgE/IgG-mediated food hypersensitivity, and consistently tracing the profile of specific antibodies in the serum of the offspring. It is possible that prolonging the study duration and including a larger population of mothers/offspring may lead to discovery of more dominant effects. Moreover, the impact of probiotic supplementation in offspring requires further examination.

## Figures and Tables

**Figure 1 nutrients-12-02381-f001:**
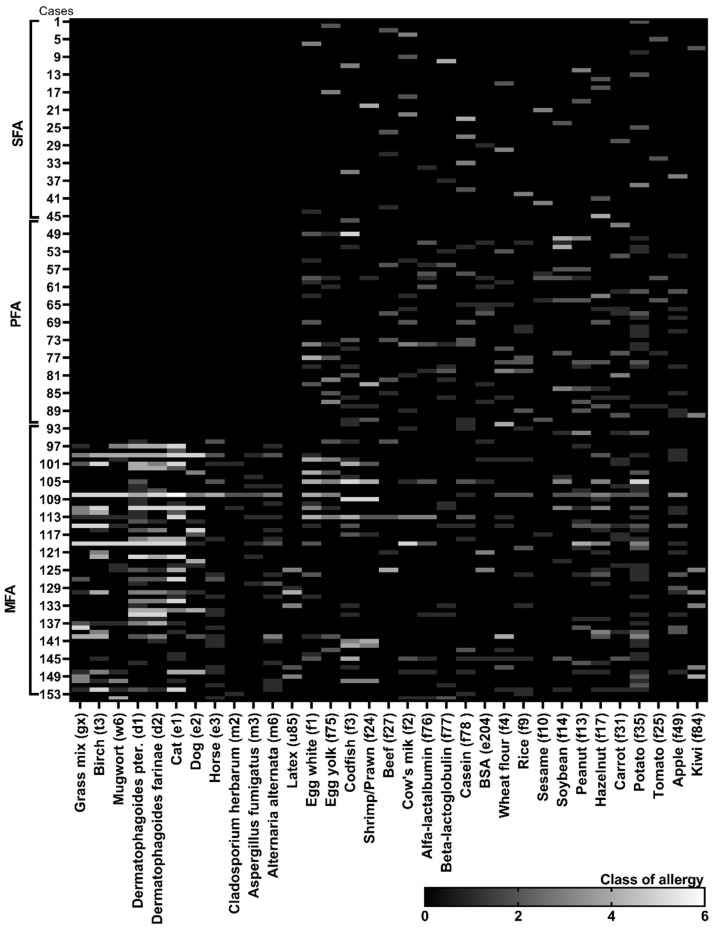
Summary of the most common food allergies and mixed polyallergies determined in characterised groups of allergic sufferers. The class of allergies was determined based on the content of specific IgE antibodies in serum. The profile and the content of specific antibodies for each tested case (*n* = 153) was presented as a class of antibody content to each antigen (from 0–6 class; allergy status from 2–6). The cases were divided to 3 subgroups: SFA—allergy to single food-origin allergen; PFA—polyallergy to various food-origin allergens; MFA—mixed polyallergy to aero- and food-origin allergens.

**Figure 2 nutrients-12-02381-f002:**
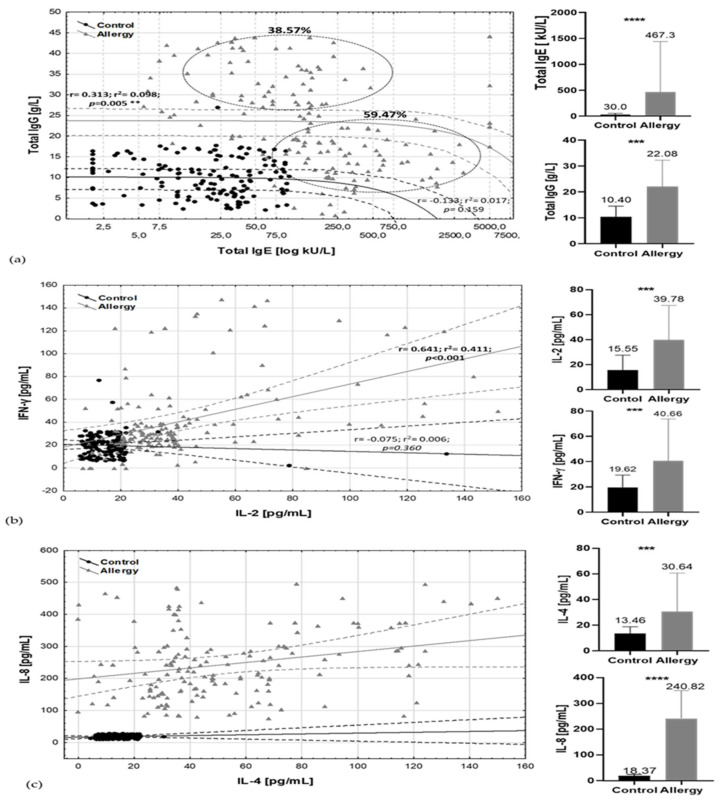
Allergic and inflammatory markers by health status categories. Spread charts include Pearson correlation parameters (*r*, *r*^2^, *p*) with 95% confidence interval and mean /standard deviation bars of major factors (**a**) total IgE/total IgG; (**b**) IL-2/IFN-γ; (**c**) IL-4/ IL-8. Black dots, lines, bars—control group; grey triangles, lines, bars—allergy group. Solid line, trend line, dashed lines, and circles: 95% confidence range. *** *t*-test, *p*-values ≤ 0.001, **** *t*-test *p*-values ≤ 0.0001.

**Figure 3 nutrients-12-02381-f003:**
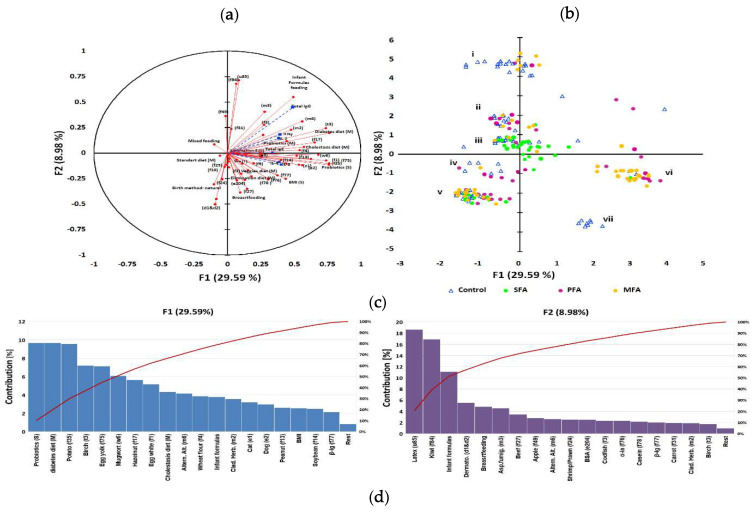
IgG level as a grouping variable in allergy profile associated to gestational and early life dietary behaviour. The principal component analysis (PCA) results: (**a**) correlation circle presenting correlations between individual specific IgE antibodies profile and mothers’ gestational and subjects’ early life dietary behaviour (red circles) with other immunological parameters as an additional variable contribution (blue squares); (**b**) score plot: circles depict allergy groups: green—SFA; purple—PFA; yellow—MFA; triangles—control, numbers i—vii refer to clusters. Further characteristic of clusters was provided in the Appendix 6. (**c**) Percent contribution of the variables in major factors, red line-contribution summing to 100%; (**d**) cluster analysis developed with an artificial neural network, illustrating the specific IgE-polyallergy profile with superimposed levels of class G antibodies and mothers’ gestational and subjects’ early life dietary behaviour. Squares indicate one or several overlapping cases, and purple squares indicate cases whose mothers declared using gynaecological probiotics during pregnancy valleys (10–30%) which gather adjacent points most similar in terms of analysed features, while mountain chains (70–100%) separate the most diverse points (cases). Letters a–g refer to the significantly clustered 95% of the most similar cases in the cluster. Further characteristics of the clusters are provided in the Appendix A.

**Figure 4 nutrients-12-02381-f004:**
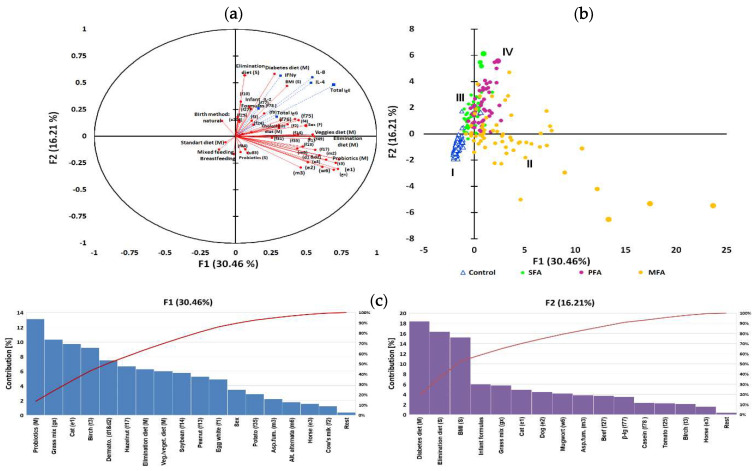
IgE level as a grouping variable in allergy profile associated to gestational and early life dietary behaviour. The principal component analysis (PCA) results: (**a**) correlation circle presenting correlations between individual specific IgE antibodies profile and mothers gestational and subjects early life dietary behaviour (red circles) with other immunological parameters as an additional variable contribution (blue squares); (**b**) score plot: circles depict allergy groups: green—SFA, purple—PFA, yellow—MFA; triangles—control, numbers I—IV refer to clusters. Further characteristics of clusters are provided in the Appendix A; (**c**) percentages of the contribution of the variables in major factors denoted as red line, contribution summing to 100%; (**d**) cluster analysis developed with an artificial neural network illustrating the specific IgE profile with superimposed levels of total E antibodies and mother’s gestational and subject’s early life dietary behaviour. Squares indicate one or several overlapping cases, and red squares indicate cases whose mothers declared who used gynaecological probiotics during pregnancy. Valleys (10–30%) gathered adjacent points most similar in terms of analysed features, while mountain chains (70–100%) separated the most diverse points (cases). Letters A–S letters refer to the significantly clustered 95% of the most similar cases in the cluster. Further characteristics of clusters are provided in the Appendix A.

**Table 1 nutrients-12-02381-t001:** Demographic and descriptive parameters of subjects, according to health status categories.

Groups by Health Status Categories Subgroups	Control	Allergy (*n* = 153)			*p*-Value ^a^
		SFA	PFAs	MFAs	
	*n* = 150	*n* = 47	*n* = 46	*n* = 60	
Age: mean (SD)	6.8 (3.1)	5.3 (2.7)	5.1 (3.9)	6.7 (3.4)	0.733 ^b^
Gender: female‒*n* (%)	77 (51.3)	28 (59.6)	30 (65.2)	36 (60.0)	0.609
Residence‒*n* (%) Urban	99 (66.0)	40 (85.1) ↑	32 (69.6)	41 (68.3)	0.012
Rural	51 (34.0)	7 (14.9) ↓	14 (30.4)	19 (31.7)	
Positive family atopy history‒*n* (%)	31 (20.7)	17 (36.2)	22 (52.2) ↑	24 (40.0) ↑	<0.001
Co-occurring chronic diseases‒*n* (%) SubjectsMothers					
	10 (6.7)	3 (6.4)	5 (10.9)	11 (18.3)	0.053
	14 (9.3)	7 (14.9)	4 (8.7)	9 (15.0)	0.489
BMI status of subjects‒*n* (%) ^c^ Underweight	16 (10.7)	11 (23.4) ↑	8 (17.4)	21 (35.0) ↑	0.006
Normal	112 (74.7)	32 (68.1)	30 (65.2)	24 (40.0) ↓	
Overweight	22 (14.6)	4 (8.5)	8 (17.4)	14 (23.3) ↑	
Motherly pregnancy dieting‒*n* (%) Elimination diet (M) Gestational diabetes diet Vegan/vegetarian diet Cholestasis low fat diet Other diet					0.032
	10 (6.7)	5 (10.6)	16 (34.8) ↑	23 (38.3) ↑	<0.001
	53 (35.3)	17 (36.2)	19 (41.3)	19 (31.7)	0.812
	19 (12.7)	1 (2.1) ↓	4 (8.7)	8 (14.0)	0.175
	7 (4.7)	3 (6.4) ↓	11 (23.9)	9 (15.0)	0.006
	5 (3.3)	0 (0.0)	1 (2.2)	0 (0.0)	0.900
Supplementation with vitamins and/or minerals‒*n* (%)	73 (48.7)	23 (48.9)	28 (60.9)	33 (55.0)	0.474
Probiotics intake‒*n* (%) Subjects Mothers					
	22 (14.6)	8 (17.0)	10 (21.7)	20 (33.3)	0.021
	19 (12.7)	12 (25.5)	16 (34.8)	25 (41.7)	0.002
Birth method: natural‒*n* (%)	97 (64.7)	32 (68.1)	25 (54.3)	30 (50.0)	0.155
Infant feeding type‒*n* (%) Exclusive breastfeeding	63 (42.0)	26 (55.3)	28 (60.9)	22 (36.6)	0.039
Formulas	41 (27.3)	13 (27.7)	7 (15.2)	13 (21.7)	
Mixed breastfeeding and formulas	46 (30.7)	8 (17.0) ↓	11 (23.9)	25 (41.7) ↑	
Elimination diet: subjects (S) *n* (%)	1 (0.7)	33 (70.2) ↑	29 (63.1) ↑	19 (31.6)	<0.001
Single food allergen	1 (0.7)	28 (59.6) ↑	11 (23.9)	5 (8.3)	<0.001
Several food allergens	0 (0.0)	5 (15.2)	18 (39.2) ↑	14 (23.3) ↑	

SFA—allergy to single food-origin allergen, PFAs—polyallergy to various food-origin allergens, MFAs—mixed polyallergy to aero-and food-origin allergens. Data are presented as the means (SD) or percentages. ^a^
*p*-Value of the chi-square test for comparison among group. ^b^
*p*-Value of the Kruskal–Wallis test for quantitative data for comparison between groups with Bonferroni’s corrected significance threshold of 0.008. ^c^ Classified according to the WHO classification [21,22]. S—subject, M—maternal. Significant associations are marked in bold. Arrows: ↑ significantly more often than in control, ↓ significantly less often than in control.

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
