# Peer review of "Gestational Dysfunction-Driven Diets and Probiotic Supplementation Correlate with the Profile of Allergen-Specific Antibodies in the Serum of Allergy Sufferers"

_nutrients, 2020, doi:10.3390/nu12082381_

Round 1
Reviewer 1 Report
Although this topic is of high interest, and the collection and analysis of data is extensive, there are several severe drawbacks in the study design:
- It is not clear if only food allergic children or children with all sorts of allergic disease are included. furthermore, lines 129/130 and 145/146: regarding food allergy: the diagnosis of allergy is based on sensitisation and not on oral food challenge tests. Sensitisation does not mean that there is clinical relevant food allergy
- the methods of collection of dietary data are not clear. In lines 160-162 the authors refer to the Europrevall semi-quantitative FFQs, however, it is not clear how detailed and extensive these were.
- the authors very much focus on IgG, however, IgG is an indicator of tolerance and corresponds to the level of consumption. Thus, for me not sure if IgG is useful in this analysis.
In general, the english is poor and the goal of the manuscript is not immediately clear from the title. The introduction and disucssion are way too long.
It is a pity, as the approach and the topic are highly interesting and promising, if a. a better defined study population was used, and b. dietary data were collected more in detail.
Author Response
Review 1
- It is not clear if only food allergic children or children with all sorts of allergic disease are included. furthermore, lines 129/130 and 145/146: regarding food allergy: the diagnosis of allergy is based on sensitisation and not on oral food challenge tests. Sensitisation does not mean that there is clinical relevant food allergy
This project was focused on children with food allergies but in the course of studies (long term 2010-2014) it occurred that in the tested group pure food allergies were not as common as mixed types. Some of the patients from the group included as pure food also in the course of the project revealed mixed types later. Patients with atopic dermatitis, asthma and isolated contact and inhalational allergies were not included in the project.
Many other factors were tested than those published here but:
- Recruitment was performed by collaborating medical doctors. Description was added in Line 154. Oral Food Challenge was done by clinicians during the phase prior the project. A collaboration with clinicians in the project was the result of their previous work in hospital but data obtained by them were not covered by the project framework and were not covered by the consent of the ethics committee agreement according this project, so the results of that should not be published here. Oral provocation was carried out in hospital conditions in the term of previous interventional studies, whereas the project group was created based on that knowledge- it was no longer necessary to repeat the provocation for the needs of the observational studies planned in this project. An agreement of the ethics committee on the procedures carried out in ambulatory terms were the decisive condition here in terms of the methods and results used for this publication.
- The measurability (precise content of antibody), unification, and the possibility to repeat the tests in the course of long distance project were crucial in this study and determined the chosen methods-considered nonetheless as acceptable alternatives for OFC in continuation of food allergy studies.
- Although the physician-supervised OFC remains the gold standard for food allergy diagnosis, a careful medical history paired with SPT and serum food-specific IgE testing often can provide a reliable diagnosis and for sure are recommended to complete the clinical picture after provocation.
Nevertheless the precise information of testing (including provocation) has been uploaded to manuscript.
- the methods of collection of dietary data are not clear. In lines 160-162 the authors refer to the Europrevall semi-quantitative FFQs, however, it is not clear how detailed and extensive these were.
Yes thank you for this comment. In fact, too big a generalization has been applied here to make the manuscript concise.
- The provided to mother FFQ questionnaire was a standardized and accurately translated questionnaire, of Harvard T.H. Chan School of Public Health Nutrition. We used the 80out formula published in 2007. The project started in 2010 and that protocol was validated and concise at the same time in that time. It includes 133 food items with specified serving sizes that are described using natural portions in standard weight and volume measures of the servings. For each food item, participants indicated their average frequency of consumption over the 9 months of pregnancy in terms of the specified serving size by checking one of nine frequency categories. (Appendix B).
- The Europreval query was only enriched with mentioned additional queries. Europreval allergy query used as a method of collection data about the course of allergy, including questions regarding family and personal history of atopy. That query included questions about birth method, infant feeding type, probiotic supplementation, current dietary habits, detailed information of the severity of manifestation induced by food intake, applied therapies of allergy, parental smoking and education, diagnosed additional diseases and psychological symptoms.
- Two additional queries were completed during the anamnesis: SCORAD scale (due to the skin manifestation severity and its connection with research on mutations in filaggrin genes). Query about taking antibiotics, non-steroidal medicines and taking probiotics by mothers and patients (in term of intestinal microbiota studies).
- In follow-up phase additional queries dedicated to eating were made SCOFF and EAT-8. Results of those queries were already published.
The dietary data were collected more in detail but our intention was to show that no one consumed only one element from the diet and plenty of data published before (cited in this manuscript) describes such relations (particular component of the diet – health condition/influence on offspring). Our aim was to check how the full pattern (qualified as a diabetic or cholestatic diet – surprising that there has been no pattern as diabetic or cholestatic set yet) affected the antibody profile in the offspring.
Both the FFQ questionnaire and results of dietary components and frequency of its consumption will be provided to the manuscript as appendices B and D, respectively.
- The authors very much focus on IgG, however, IgG is an indicator of tolerance and corresponds to the level of consumption. Thus, for me not sure if IgG is useful in this analysis.
Yes thank you for this comment. The focus on total IgG was caused by the fact that we determined the whole picture of processes not only allergy development and progress but also the influence on tolerance development and IgG mediated inflammatory diseases too. The profile of specific antibodies were the most important though. Total IgE/ IgG were only factors helpful in discrimination of patients subgroups. We used all the time the panels of tests for specific IgE antibodies in the course of allergy describing. IgG was considered as a covariant, not as a major factor of allergy discrimination. The aim was to test the influence of tested patterns, components and probiotics on allergy/tolerance development, pointing out which pattern may predestine the further course of allergy and specific content and profile of antibodies in serum. That was also highlighted in discussion. Also refereed paper was describing similar observation of gestational desensitization through IgG1 and IgG4 translocation.
It was cited in our paper as follows:
All the mentioned results suggest maternal allergen-specific IgG antibodies’ protective influence on offspring which was also observed in the context of gestational allergen-specific immunotherapy (AIT), and placental transfer of allergen specific IgG1 and IgG4 antibodies, but not IgE, from AIT-treated mothers to offspring. For 164 tested allergens, children from mothers with increased (>30 ISAC standardized units) specific plasma IgG levels against an allergen developed no IgE sensitizations against that allergen at 5 years of age [38].
Authors know that many different papers describe the tolerogenic properties of IgG in the context of allergy. In fact, higher levels of IgG4 to foods may simply be associated with tolerance to those foods and in the course of immunotherapy its proportion may reach even 75% of IgG total. It should be also related with the other parameters because the flexibility of the hinge region in IgG is pretty impressive. In tested group IL-10 was really low as well FoxP3 in cell populations but IL-2, IL-8 and TNFα were impressive so that is why the levels of those were presented as additional tested parameters. Authors mentioned in this study about some different types of diseases in the course of which IgG secretion and its role on Fcγ receptors might be also important. Mentioned common genetic background of autoimmunological diseases (diabetes type 1 and enteropathy –commonly co-occurring in the course of atopic diseases). The presentation of total IgG content was according to the authors justified. And that is why authors cautiously use phrases regarding allergies in the context of IgG.

Reviewer 2 Report
Gestational dysfunction-driven diets and probiotic supplementation correlate with the profile of allergen-specific antibodies in the serum of allergy sufferers
The paper by Ogrodowczyk et al. examines the role of maternal diet during gestation on childhood allergies. The research questions being asked are very important and experiments are well executed. However, I have substantial concerns with the multivariate analyses. As a whole there is a paucity of statistical support for the conclusions drawn from the multivariate analyses. A central argument of the paper is that an artificial neural network was able to detect a signal across the dataset. However, there is so little detail regarding the methods of the ESOM presented in the manuscript that it is impossible to evaluate these results and any conclusions that can be drawn from them. It is unclear how these results were validated or how models were evaluated. Neural networks are very good at finding signals in data but are highly susceptible to overfitting the data. Because there is no description of how the models were trained and evaluated, it remains unclear how much these results will be generalizable.
Major Concerns:
As stated above, It is not clear how the analysis has been done throughout the paper. The reasoning behind the various types of analyses and the methods chosen are also not clear. I have already stated my concerns on the Databionic ESOM analyses. The crux of the paper, which is the result of this neural network, has minimal interpretability. The problem with approaches such as this is that they prioritize finding a signal in the data at the expense of inferring the relationship. So, these data suggest that there is some signal between various factors, but it remains unclear how these factors are related and how they respond to fluctuations in one another. My recommendation would be to use an L1 penalized logistic regression model to identify covariates related to your various allergy classes then use a simpler, yet more interpretable, model to elucidate the relationship between the variables identified via the penalized logistic regression model and your allergy classes. That would offer greater support for the conclusions of the paper which, at the moment, are not strongly supported.
The figures we get from this analyses are also blurry and the reader strains to understand the data. In addition, authors mention that they used parametric t test for their analysis. It is very unusual for biological data to be normally distributed and non-parametric t test would be better suited.
In summary, the entire data should be reanalyzed with better analytical tools. I consider this the most important aspect that needs to be worked on before the manuscript can be published.
If there is an advantage to the ESOM approach that this reviewer is missing then there must be a substantial expansion of the methodology surrounding the model. How was it validated? Was error measured? Was the model optimized? This is a tremendous amount of work that should be considered essential if this is to remain in the manuscript. The code underlying these analyses must also be made publicly available.
Secondly, there are numerous grammatical errors throughout the paper and I could not highlight all of them. This makes the understanding of the manuscript difficult. Authors should reread and edit the paper in detail.
Other Concerns
Introduction
Line 52: Insert reference
Line 59: “In Poland, in 2016, it was even 8%”. Combine this with the previous sentence
Line 63: Correct the punctuation in this sentence
Line 74: Authors should cite the studies that have discussed fetal programming as a means of preventing atopic conditions
“It gives hope for exploring the interaction between maternal dietary patterns and genetic factors to suggest dietary changes that could prevent or reduce hypersensitivity with significant reduction in IgG-dependent reactions [12] and, what still remains a challenge, IgE-reactions”
Rewrite this sentence. It is confusing as written and not clear what the message is.
Line 77: This sentence is hanging. Rewrite including reference or remove altogether
Line 77-80: To better convey the message, please break this into two sentences and include the correct punctuation. There is a severe failure in oral tolerance early in life. It is not clear whether the this outcome improves or worsens later in adulthood from the subsequent sentence.
Line 81: Please reference
Line 83: Instead of “confirmed”, use “demonstrated” or “shown”. Also, these are studies that have been done by others. Please recognize their work by properly citing. This entire sentence is not cited.
Line 90:What does this sentence mean?
Line 99: The paper by Grieger et al looks at various types of allergies. Specify this in text instead of saying “each allergy”. Be specific here.
Line 99: What is FFQs? This is the first time you are using it
Line 104-105: I am not sure I understand what gestational dysfunction means here? Please make this more clear. Do the authors means variations in diet during gestation or changing dietary habits during gestation? Not clear to me
Line 111: See comments above on dysfunction
Line 113-117: Use concise sentences. It is difficult to follow the way currently written.
Materials and Methods
Line 125: Are these infants with allergies? Can you specify which allergies please. Grammar is also poor
Line 129: Change to serum IgE instead
Line 130: Which inflammatory cytokines?
Line 146: Provide the reference for this cut off for specific IgE
Line 155: Change INF?
General comment: At what point was all the immunological analyses done? When was blood collected to look at Ig levels and cytokines?
Authors should specify
Assessment section: Please include the questionnaire that was used as a supplementary figure
Line 166-167: What do authors mean by “acceptance of rules?” Please rephrase this sentence
Lines 169-178: This needs drastically expanded. What parameters were used? What data were used? What specific algorithm was used to construct the map? It’s hard to understand the Figs from this analyses. I would recommend the authors use a different analyses method or explain this better to the reader if they really have to use it.
Lines 188-189: How did you arrive at this threshold?
Line 194-196:References for this analyses
Line 198-200: “Quantitative supplementary data were immunological parameters”. This sentence is confusing
Results:
Line 206-208: Equal subsets of the 153?Then that should be 50% and not 31% as currently stated. Once again, the grammar should be improved
Fig 1: The quality of this heatmap is very poor. This should be improved for clarity or better still, the data should be shown in a table. Is this the diet of the infants or what the mothers had during gestation? This should be clearly spelled out in the figure legend. For the classes of allergy, we have 0,2,4 and 6. Do this denote anything on the heatmap? The intensity of shade should match an allergy type? It’s difficult to understand as presently shown. This should be improved or shown in a different format.
Table 1: Values in the table should be aligned to match the different headlines
Line 210-211: What does this sentence mean?
Line 211: “Slight tendency” change this statement. If there is no difference, state it clearly. There is nothing wrong in there being no difference.
Line 226: What is gestational elimination diet? How was this defined?
Line 227-228: Was this correlation done in the controls, cases or all study participants? What was the % of control mothers with chronic infections?
Line 229: Cite the data and include p values
Line 234-235: Where is this data? How did you measure diet adherence? What does diet adherence mean?
Line 235-237: What does this sentence mean? Where is the IgE data being referred to here? First, authors are correlating severity of allergy to diet adherence which I don’t (data which I can’t see), secondly, assessing severity based on IgE, again we don’t have reference to that data in text.
Line 241: What is “even two gestational dysfunction diet?” I have commented on this already, this needs to be defined for the reader
Line 250-253: Reference the data
Line 285: What does the r value represent here? I’m assuming the authors looked at IgE levels in males versus females? If that’s the case, then a simple t test is sufficient to check the stats?
Line 286: What is the significance of the IgE/IgG ratio in this context?
Fig 2A:
Do the circled clusters correlate with any other covariates?
What are the concentrations depicted for the antibody?
Fig 2 Generally: For the bar plots, the second group is defined as “Allergy’ in some plots and “Allergic” in others. Be consistent.
Line 290-291: This sentence doesn’t mean anything
Line 296-297: Where is this data?
Line 323-325: rewrite this sentence please
Line 333-337: The text doesn’t seem to match what is shown in Fig 3A. First, which graph shows the variability described? Secondly, which 9 parameters are these? Writing should be made clear to capture the results being described. The data should also be shown. Looks like authors are describing data that is not shown here.
Figure 3: so is Figure 3a a constrained analysis of principal components? The figure title suggests as much.
Figure 3b: it feels like all of the drawn ellipsoids are leading the reader, I would recommend removal. I’m not sure that these clusters really correlate strongly with the different allergy types, so I’m not sure what they contribute to the story. At the very least, we need to see statistical support for the validity of these clusters.
Fig 3c: Very difficult to read what is in the X axis. Use Arial at least font 11. Relevance of the redline not clear
Fig 3d: This analysis method is difficult to follow. As I highlighted earlier, authors should provide justification for using this method as the ellipsoids feel like they’re guiding the reader. The resolution is also prohibitive for reading.
Looking at figure 3, these clusters look somewhat artificial and don’t seem to correlate with any of the allergy covariates discussed. I would recommend performing an Adonis to assess if the intra-group distance if significantly lower that the inter-group distance.
Section 3.7
Line 15-16: How did authors arrive at this statement? What adjustment? If this is the basis for the analyses in this section, then it must be clearly described.
Figure 4b: again, the ellipsoids seem to be guiding the reader. The recommendation for performing an Adonis holds for this as well.
Figure 4d: again, the clusters seem to guide the reader and don’t appear to be supported by the data. This figure is not convincing nor easily interpretable.
Author Response
Reviewer 2
Thank you for your insightful and constructive review. Responses to most of the guidelines were uploaded directly to the manuscript but also some of them were commented here. The English in the manuscript has been checked by a native English speaking person.
In the opinion of the authors, the role of the neural network as a method indicating possible associations that might be later validated is important. In this study there were various methods of data analysis tested with varying degrees of restrictiveness. They invariably showed the relationships reported in this manuscript. The authors are aware that both the method and the size of the studied population may be important here. As Reviewer mentioned, the neural networks are very good at finding signals in data but are highly susceptible to overfitting the data and actually the authors were most interested in these properties of the network. The authors are aware of this phenomenon, but at the same time they are aware that the studied population was not very large, so these studies were treated rather as a pilot in this manner, and both the method and tools will be more thoroughly verified in the future and at that time it will be possible to test how much these results will be generalizable. The authors do not deny that they will use the suggested analysis L1 penalized logistic regression then. Due to the fact that the consent of the ethics committee did not provide for continuation of the research, which also applies to the consent to re-analyze the data, now the analysis cannot be performed. On the other hand, a double analysis of phenomena was provided to the manuscript, also using the more popular methods of multivariate analyzes. The authors agree that more detailed descriptions of both the ESOM tool and the presentation of the statistical support of the results are needed. Some suggested values were added to the manuscript. In the original version, they were omitted because the work was supposed to be more biological, not purely methodical. The introduction of a large number of statistical values seemed to the authors to complicate the merits they wanted to present in a simple way, which is not easy for such a large number of analyzed variables. The authors agree that strength of some associationes might seem not strong but it were still relevant that is why it was reported and concluded like that. Unfortunately, neural maps are generated in the best possible resolution. Because of a small number of cases (only 303), the maps are generated of small sizes. It is only possible to fold the image so that the data might be multiplied and has no borders. Due to the low legibility of the numbers representing specific patient cases, they have been removed because they do not add much to the manuscript and the examination of individual cases also seems pointless if some general regularities are sought. Hopefully this will make the maps clearer to the Reviewer.
Moreover, in the research conducted, most of the time a comparison of 4 groups is used - 3 allergic groups and one control group. The use of mentioned parametric t test in the research concerned only one comparison in which the mean age proportion between all allergic sufferers and control was tested. The studied population, despite the lack of typical randomization of studies, had a normal distribution of data and, surprisingly, equal variance and only because of this the parametric test was used. This situation changed when the allergy group was divided into 3 subgroups. Significance there has been tested using Chi-Square test and results were presented in Table 1.
Introduction
The correction for comments to Lines 52-63 had been uploaded to the manuscript.
Line 74: Authors should cite the studies that have discussed fetal programming as a means of preventing atopic conditions
In the authors' opinion now this is more clear:
The role of fetal programming in the prevention and therapy of various inflammatory and atopic diseases is also widely discussed. Randomized controlled trials on the influence of dietary intervention on epigenetic mechanisms in children suffering from cow’s milk allergy had been already done [12]. The maternal dietary patterns may prevent or reduce hypersensitivity with significant reduction especially in IgG-mediated diseases [13]. A successful protocol for IgE-mediated reactions still remains a challenge but there have been published several studies describing correlations between maternal dietary components that works preventively through epigenetic activation of mechanisms against asthma, wheezing and allergic rhinitis [14].
The correction for comments to Lines 77-90 had been uploaded to the manuscript.
Line 99: The paper by Grieger et al looks at various types of allergies. Specify this in text instead of saying “each allergy”. Be specific here.
Actually the meaning of ‘each allergy’ was different than it is typically used because that statement was used in the context of a combination of atopic diseases like eczema, current wheeze, or rhinitis but not citing this precisely like the authors did we used the phrase ‘each allergy’. It has been better explained in the manuscript.
The correction for comments to Line 99 and Line 113-117 has been uploaded to the manuscript.
Line 104-105, and Line 111. I am not sure I understand what gestational dysfunction means here? Please make this more clear. Do the authors means variations in diet during gestation or changing dietary habits during gestation? Not clear to me
Authors meant that it is health dysfunction appearing during pregnancy (like gestational diabetes, cholestasis) that appear during pregnancy and usually disappear after childbirth, but require dietary adjustment during pregnancy. That has been clarified in the text.
Materials and Methods
Line 125: Are these infants with allergies? Can you specify which allergies please.
Mentioned project was focused on children with food allergies but in the course of studies (long term 2010-2014) it occurred that in the tested group pure food allergies were not as common as mixed types and some of the patients from the group included as pure food also in the course of the project revealed mixed types later (mixed means food with pollen or latex). Patients with isolated atopic dermatitis, asthma and contact and inhalational allergies, who had no exacerbation of symptoms after any food consumption, and who had no specific IgE to food in serum were not included in the project. We do not call an allergy symptoms specific to atopic dermatitis and to asthma like it has been cited above by Grieger et al. It has been emphasized in manuscript.
The correction for comments to Lines 129-155 had been uploaded to the manuscript.
General comment: At what point was all the immunological analyses done? When was blood collected to look at Ig levels and cytokines?
This part has been described in detail in a previously published paper cited in this manuscript. Line 142. ‘All the protocols have already been described in the authors' publications’.
But we specified this paragraph:
‘Briefly: blood and serum samples were collected twice during the project period. Once during the recruitment phase and again after 5 years. All the immunological analyses were carried out using tests certified for clinical trials with declared norms for human samples and certified laboratories.
Assessment section: Please include the questionnaire that was used as a supplementary figure.
The queries prepared for this study were prepared in Polish language. Moreover, they were very spacious. The translated questionnaires were included to manuscript as Appendix B and C but due to their length it cannot be a figure. They are provided in a protected PDF file format.
Line 166-167: What do authors mean by “acceptance of rules?” Please rephrase this sentence
The formula required by the ethics committee that provided the acceptance for this study. In the formula provided to the committee there were specified special rules of project for the participants and the document called ‘acceptance of the rule’ has been provided to participants. The special rule of this project was: -repeated contact for re-assembly of biological material (fecal sample, urine, blood) and surveys after the phase of confirmation the presence of specific IgE antibodies in serum.
It was described in the quoted publication but for the need of this manuscript more detailed information has been provided to the manuscript.
Lines 169-178: This needs drastically expanded. What parameters were used? What data were used? What specific algorithm was used to construct the map? It’s hard to understand the Figs from this analyses. I would recommend the authors use a different analyses method or explain this better to the reader if they really have to use it.
The section was extended with methodological parameters of the performed analysis including data pre-processing, training process and the testing method.
Lines 188-189: How did you arrive at this threshold?
ᾱ =0.008 value was computed for the number of test m=6 calculated for 4 compared groups. We have been comparing 4 groups in our test. Control vs 3 types of allergies (SFA, PFA, MFA).
The correction for comments to Lines 194-196 and 198-200 had been uploaded to the manuscript.
Results:
Line 206-208: Equal subsets of the 153?Then that should be 50% and not 31% as currently stated. Once again, the grammar should be improved
That would be true if we were comparing 2 groups, not 3. But we compared 3 different food allergic types. For 3 groups about 30% is equal- but to avoid misunderstanding it has been corrected.
Fig 1: The quality of this heatmap is very poor. This should be improved for clarity or better still, the data should be shown in a table. Is this the diet of the infants or what the mothers had during gestation? This should be clearly spelled out in the figure legend. For the classes of allergy, we have 0,2,4 and 6. Do this denote anything on the heatmap? The intensity of shade should match an allergy type? It’s difficult to understand as presently shown. This should be improved or shown in a different format.
Yes thank you for the comment but provided in Fig.1 data describe profiles of specific IgE antibodies profile and content in the serum of each of 153 allergic participant. There were no data describing any diet here. The entire publication essentially emphasizes the importance of documenting a specific serum antibody profile. The content of antibodies or severity of allergy are very often presented as a class of allergy in a scale 0-6 or sometimes as 0-7 when 7 is an anaphylaxis. We corrected the explanation in the legend. The color of the point on the heatmap is important then because that informs how intensive is the response of the patient to the allergen. Based on this and the EuroPrevall Project comments, the cut-off 0.7kU/L was established and the results of this test were presented as classes.
Class of allergy |
Specific IgE content kU/L |
Severity of manifestation |
1 |
0-0.35 |
no specific antibodies, no manifestation |
2 |
0.35-0.7 |
low content of specific antibodies, very mild manifestation |
3 |
0.7-3.5 |
low content of specific antibodies, mild manifestation |
4 |
3.5-17.5 |
medium content of specific antibodies with moderate manifestation |
5 |
17.5 - 50.0 |
high content of specific antibodies with manifestation |
6 |
50.0-100.00 |
very high content of specific antibodies with severe manifestation |
The presentation of the results in a table would be difficult to understand and hardly visible because it is a table with 154 rows and 35 columns which basically does not give us a picture of the intensity of the manifestation in such a readable way as a self-reading heatmap. The quality of the map on the computer seems good and legible.
We will provide the excel file with tabular data if you want to compare (Excel file: class and type of allergy table).
Table 1: Values in the table should be aligned to match the different headlines
Yes we agree that the table changed the formatting. For some reason the files sent by us on the editor's templates have been reformatted, but we don't know why it happened. In PDF it was ok but not sure what exactly happened and we are not sure then what you mean in Table 1 in this situation because it changed totally. It also cut a few of the last lines of the results. That is the reason why this table has been edited and uploaded in this format here.
Line 210-211: What does this sentence mean?
It means that the participants were included based on their health status not according to the rule of random selection with the criterion of demographic equality. Nevertheless both groups were comparable in terms of age and equal sex participation was maintained in the study.
Line 211: “Slight tendency” change this statement. If there is no difference, state it clearly. There is nothing wrong in there being no difference.
We agree so this information was delated in the manuscript.
Line 226: What is gestational elimination diet? How was this defined?
As it was described above in the answer for suggestion to Line 104-105, and Line 111 the information about diets were provided in paragraph introduction.
The gestational elimination diet was declared by mothers with their positive history of allergy / intolerance. That diet was considered to be preventive and assumed an elimination of strong allergens like: milk, wheat, nuts.
Please note that data were collected and described the period of pregnancy that was even earlier than 2010-2014. At that time the elimination diet was still a commonly used protocol in pregnancy. Its legitimacy has recently been questioned and the EACCI has issued a guideline to stop this practice.
Line 227-228: Was this correlation done in the controls, cases or all study participants? What was the % of control mothers with chronic infections?
Chronic diseases are not infectious in this study. The percentage share of mothers in each tested group was provided in Table 1.
Control: 9.3%, SFA: 14.9%, PFA: 8.7%, MFA: 15%. No significant difference was stated p=0.489. The correlation was done in all study participants then.
Line 229: Cite the data and include p values.
The data were also provided in Table 1 but of course we referred them in the text.
Line 234-235: Where is this data? How did you measure diet adherence? What does diet adherence mean?
In the provided FFQ one of the questions was about scrupulousness in following the diet. The criterion measured was the number of deviations from the diet during the week.
In the case of special diets, such as diabetic or cholestatic diets, the neglect of which may cause more severe complications, adherence to the diet was high, meaning less deviations.
This single parameter has been only provided in the text. The authors did not attach additional extensive tables with the results of the FFQ because, in their opinion no one consumes only one element from the diet and data published before (cited in this manuscript) describes such relations (particular component of the diet – health condition/influence on offspring). Our aim was to check how the full pattern (qualified as a diabetic /cholestatic/ plant based/ elimination diet) affected the antibody profile in the offspring. Because many publications already referred to this type of data, here the authors wanted to focus only on patterns and their relation to IgE profile. Nevertheless, these data will be included as Appendix D.
Line 235-237: What does this sentence mean? Where is the IgE data being referred to here? First, authors are correlating severity of allergy to diet adherence which I don’t (data which I can’t see), secondly, assessing severity based on IgE, again we don’t have reference to that data in text.
The severity of allergy manifestation was measured based on self-reports of patients in EuroPrevall questionnaire on 10 points interval scale. Where 1-is not severe, almost imperceptible and 10-is extremely severe, hard to handle. It has been compared with the content (class) of specific IgE antibody in each case.
Referring to the IgE data that were provided in Fig.1: The correlation was made for each specific class of antibody for specific allergen in each case and severity so, as you can imagine, the number of correlated and compared factors is large (in the vast majority of correlated parameters no correlation was observed, so that sentence refers to this observation) and not all negative ones were shown. Only those significant are commented on in the text. It was correlation of described above factors Q1: severity with Q2: the declared adherence to diet. It was the only observed positive effect of this analysis. Mentioned in text as: The severity of allergy manifestation was moderately correlated with the tendency towards diets adherence (r = 0.489; p = 0.005).
However, it loses its potency when the specific IgE level criterion is added:
The sentence: Nevertheless it has shown no dependence of severity of allergy symptoms and the type of allergy and subjects’ serum IgE level. The severity appeared to be strictly individual tendencies.
Line 241: What is “even two gestational dysfunction diet?” I have commented on this already, this needs to be defined for the reader
Yes, thank you, all the diets have been specified in the introduction.
Following two gestational dysfunction diets means that diets are not mutually exclusive. Adherence to one, e.g., low-cholesterol, was often declared with an additional diabetic diet, and the diabetic one was often declared with a vegetarian diet.
Line 250-253: Reference the data
The reference was added. It was referred above in Table 1 and now Appendix D.
Line 285: What does the r value represent here? I’m assuming the authors looked at IgE levels in males versus females? If that’s the case, then a simple t test is sufficient to check the stats?
Actually it was a combined factor Q1: sex and Q2: type and strength of allergy. So the mixed one let us to check if the gender had any real influence. It was observed in the later mentioned PCA analysis and neural maps so this is the effect of validating this phenomenon but described in a way that seemed to be suitable for reporting this.
Line 286: What is the significance of the IgE/IgG ratio in this context?
The focus on total IgG was caused by the fact we determined the whole picture of processes not only allergy development and progress but also the influence on tolerance development and IgG mediated inflammatory diseases. The profile of specific antibodies were the most important though. Total IgE/ IgG were only factors helpful in discrimination of patients’ subgroups. We always used the panels of tests for specific IgE antibodies in the course of allergy description. IgG was considered as a covariant, not as a major factor of allergy discrimination. Authors mentioned in this study about some different types of diseases in the course of which IgG secretion and its role on Fcγ receptors might be also important. Mentioned common genetic background of autoimmunological diseases (diabetes type 1 and enteropathy –commonly co-occurring in the course of atopic diseases). The presentation of total IgG content and discriminant IgE/IgG was, according to the authors, justified in the context of the direction of changes taking place in offspring.
Fig 2A:
Do the circled clusters correlate with any other covariates?
The r-factor and other parameters are determined for allergic vs. control groups, so refer to the trend line with 95% confidence interval and not for clusters marked with ellipses. The ellipses group the cases of the highest homogeneity in terms of the examined features: IgE and IgG content.
What are the concentrations depicted for the antibody? Fig 2 Generally: For the bar plots, the second group is defined as “Allergy’ in some plots and “Allergic” in others. Be consisten
The correction of units and unification of descriptions on the graphic has been made.
Line 290-291: This sentence doesn’t mean anything
That is true. The sentence has been specified
Line 296-297: Where is this data?
In table 1 currently. Presented only as p- value. Results for BMI and blood parameters abnormalities that were listed at the end of the Table 1 but were cut off when the table was reformatted now are provided once more.
Line 323-325: Rewrite this sentence please
The sentence has been rewritten and specified.
Line 333-337: The text doesn’t seem to match what is shown in Fig 3A. First, which graph shows the variability described? Secondly, which 9 parameters are these? Writing should be made clear to capture the results being described. The data should also be shown. Looks like authors are describing data that is not shown here.
The description has been corrected. The provided description mostly referred to method of typing the major adjusting factor (contents of IgE and IgG) for PCA analysis. It described the role of two major factors F1 and 2. The left 7 F3-9 has been provided in Appendix 8. At the same time, authors’ attempts were to avoid giving the chapter titles like ‘PCA analysis of results.’
Once response for comments of Reviewer about Figure 3b, 3d, 4b and 4d will be provided because the way in which all statistical data were compiled and validated in the presented results was analogous in all cases of the discussed clusters emerged in multivariate analyses:
The coherence of the clusters has been tested. Dunn test was used to estimate the average distance between clusters. Only significant clusters have been circled for easier reading of the illustrations. All the clusters were also tested in terms of the strength of multiple correlation/regression of tested data. Those tests were done to check the strength of association of tested variable with the set of other grouping variables in each cluster. The coefficient of multiple correlation takes values between 0.00 and 1.00. On this basis, it was possible to infer the strength but not the direction of changes. It has been presented in a table. It allowed, however, to assess the direction for further research and for testing individual interactions using the Pearson correlation. On this basis, the authors concluded about the contribution of individual factors and the direction of changes. This approach seemed clearer than giving r and p for each cluster in the illustration. The results were then summarized in tables and are presented in the appendix 9. The authors wanted a simple presentation of the results so that they would be legible not only to statistics specialists. Therefore, simple descriptions and forms of presentation were used with ellipsoids. We agree that it looked like an important grouping factor in PCA analyses was the type of allergy but such factor was not tested as a grouping factor. Such grouping occurred spontaneously. The authors do not see anything wrong with it, because it is necessary to interpret the data on the influence of diet on the allergy profile and specific antibodies on some basis, which was actually the aim of this publication. At the reviewer's suggestion, the ellipsoids were removed and replaced with a letter designation corresponding to the results in the table in appendixes 5,6 and 9.
Fig 3c: Very difficult to read what is in the X axis. Use Arial at least font 11. Relevance of the redline not clear
The used font is now even 14 but its adjustment to the page size A4 caused that it becomes smaller. On PC screen it is readable, and with such an amount of data, the font apparently compresses itself when is printed despite embedding fonts.
Section 3.7
Line 15-16: How did authors arrive at this statement? What adjustment? If this is the basis for the analyses in this section, then it must be clearly described.
This adjustment was described in section 3.6 and is based on the results presented in Figure 2a. The appropriate explanation has been added.

Round 2
Reviewer 1 Report
thank you for your response.
to my first question: furthermore, lines 129/130 and 145/146: regarding food allergy: the diagnosis of allergy is based on sensitisation and not on oral food challenge tests....
you state that in line 154 you have added a sentence about diagnosis in earlier collaboration by the physician, however, I cannot find this sentence.
instead: in lines 140-145 it says: All the participants were previously patients of collaborating allergy clinics. 153 allergy sufferers and 150 healthy individuals together with mothers (114 and 108, respectively) were included. Inclusion criteria for food allergy patients were in accordance with
European Society for Pediatric Gastroenterology, Hepatology and Nutrition – ESPGHAN 2015.
Briefly: 1) manifestations specific for food allergic disease; 2) confirmed status by immunological tests, with increased levels of total and specific serum IgE and cytokines (IL-2, IL-4, 144 IL-8, IFN-γ) ; 3) positive results of percutaneous skin tests; 4) known allergy family history.
thus, this study is based on positive sensitisation and not on oral food challenge tests.
Thus, in the title the word "Allergy sufferers" should be replaced to "sensitised women"
Author Response
Review 1 round 2
Thank you so much for this comments.
If what I am writing still seems unclear then I will try to explain it again using graph.
Explanation for the coment:
To my first question: furthermore, lines 129/130 and 145/146: regarding food allergy: the diagnosis of allergy is based on sensitisation and not on oral food challenge tests....you state that in line 154 you have added a sentence about diagnosis in earlier collaboration by the physician, however, I cannot find this sentence.Instead: in lines 140-145 it says: All the participants were previously patients of collaborating allergy clinics. 153 allergy sufferers and 150 healthy individuals together with mothers (114 and 108, respectively) were included. Inclusion criteria for food allergy patients were in accordance withEuropean Society for Pediatric Gastroenterology, Hepatology and Nutrition – ESPGHAN 2015.Briefly: 1) manifestations specific for food allergic disease; 2) confirmed status by immunological tests, with increased levels of total and specific serum IgE and cytokines (IL-2, IL-4, IL-8, IFN-γ) ; 3) positive results of percutaneous skin tests; 4) known allergy family history.
thus, this study is based on positive sensitisation and not on oral food challenge tests.
Yes thank you for this comment.
2008 We realized an increasing incidence of food allergies in the reports of allergologists. Their observations were formulated on the basis of the results of research, including provocation tests that were performed in a hospital. |
2009 W applied for funds in agendas and we raised funds and with financing in the project we applied for the approval of the ethics committee |
2010-2014 Time frames of project We collected serum from patients tested previously in 2008 by allergologists. We have been only making tests for those same people using serum tests and collaborating allergologists were confirming that with skin tests. There were no sense to make once more provocation and the ethics committee provided only acceptance for serum/skin tests. |
So then we can publish the data for which we gained acceptance of committee but it does not mean that all our knowledge about the health status of participants is based only on sensitization. Not everything we should publish though. However following your ask we mentioned about this in corrected manuscript in lines 162- 163.
Line 162-163. All the participants were previously patients of collaborating allergy clinics and the confirmation of allergy was done in in a hospital setting, including provocation tests
Explanation for suggestion:
Thus, in the title the word "Allergy sufferers" should be replaced to "sensitised women"
All the studies were dedicated to children not mothers so we cannot replace statement “Allergy sufferers" in manuscript to "sensitised women". We have been testing the impact of maternal diet and supplementation during pregnancy on sensitization of offspring. Of course some mothers were allergic too but it was not our goal.
Also I would like to inform you kindly that the English has been checked many times in this manuscript first of all by 2 native English speaking people but also by a certified editorial service Translmed No: 42272020 TYPE: L2 DATE: 26/07/2020 EDITOR: VR/TRANSL SPELLING: AmE. Not only vocabulary but also the style was revised.
I hope there will be no doubts about this manuscript more but if there are any, I will try to clarify it all.
Reviewer 2 Report
The authors have made a significant effort to improve the manuscript compared to the previous version. Various aspects that were unclear in the introduction alongside the numerous grammatical issues have been significantly improved.
I also would like to thank the authors for their detailed expansion of the methods section.
The paper can be published after authors address these minor comments.
Minor comments:
Fig 4: Line 11-correct spelling
110-111-Insert the citation correctly
201-202: Reference these studies please. Cox et al, Nobel et al., Nyangahu et al., Gomez-Aguero et al etc
203: Reference?
Author Response
Review 2/2
Thank you so much for this comments.
Manuscript has been corrected directly in the text of manuscript. Also additional references were added. I hope this time the manuscript is complete.
Fig 4: Line 11-correct spelling
Spelling mistakes has been corrected.
110-111-Insert the citation correctly
The citation has been corrected.
Review 2/2
Thank you so much for this comments.
Manuscript has been corrected directly in the text of manuscript. Also additional references were added. I hope this time the manuscript is complete.
Fig 4: Line 11-correct spelling
Spelling mistakes has been corrected.
110-111-Insert the citation correctly
The citation has been corrected.
201-202: Reference these studies please. Cox et al, Nobel et al., Nyangahu et al., Gomez-Aguero et al etc
Additional citations has been provided.
203: Reference?
The reference has been added.
201-202: Reference these studies please. Cox et al, Nobel et al., Nyangahu et al., Gomez-Aguero et al etc
Additional citations has been provided.
203: Reference?
The reference has been added.
Thank you
